# Linear Mode Connectivity in Differentiable Tree Ensembles

## Abstract

*Linear Mode Connectivity* (LMC) refers to the phenomenon that performance remains consistent for linearly interpolated models in the parameter space. For independently optimized model pairs from different random initializations, achieving LMC is considered crucial for validating the stable success of the non-convex optimization in modern machine learning models and for facilitating practical parameter-based operations such as model merging. While LMC has been achieved for neural networks by considering the permutation invariance of neurons in each hidden layer, its attainment for other models remains an open question. In this paper, we first achieve LMC for *soft tree ensembles*, which are tree-based differentiable models extensively used in practice. We show the necessity of incorporating two invariances: *subtree flip invariance* and *splitting order invariance*, which do not exist in neural networks but are inherent to tree architectures, in addition to permutation invariance of trees. Moreover, we demonstrate that it is even possible to exclude such additional invariances while keeping LMC by designing *decision list*-based tree architectures, where such invariances do not exist by definition. Our findings indicate the significance of accounting for architecture-specific invariances in achieving LMC.

## 1 Introduction

A non-trivial empirical characteristic of modern machine learning models trained using gradient methods is that models trained from different random initializations could become functionally almost equivalent, even though their parameter representations differ. If the outcomes of all training sessions converge to the same local minima, this empirical phenomenon can be understood. However, considering the complex non-convex nature of the loss surface, the optimization results are unlikely to converge to the same local minima. In recent years, particularly within the context of neural networks, the transformation of model parameters while preserving functional equivalence has been explored by considering the *permutation invariance* of neurons in each hidden layer [1, 2]. Notably, only a slight performance degradation has been observed when using weights derived through linear interpolation between permuted parameters obtained from different training processes [3, 4]. This demonstrates that the trained models reside in different, yet functionally equivalent, local minima. This situation is referred to as *Linear Mode Connectivity* (LMC) [5]. From a theoretical perspective, LMC is crucial for supporting the stable and successful application of non-convex optimization. In addition, LMC also holds significant practical importance, enabling techniques such as model merging [6, 7] by weight-space parameter averaging.

Although neural networks are most extensively studied among the models trained using gradient methods, other models also thrive in real-world applications. A representative is tree ensemble models, such as random forests [8]. While they are originally trained by not gradient but greedy algorithms, differentiable *soft tree ensembles*, which learn parameters of the entire model through gradient-based

optimization, have recently been actively studied. Not only empirical studies regarding accuracy and interpretability [9–11], but also theoretical analyses have been performed [12, 13]. Moreover, the differentiability of soft trees allows for integration with various deep learning methodologies, including fine-tuning [14], dropout [15], and various stochastic gradient descent methods [16, 17]. Furthermore, the soft tree represents the most elementary form of a hierarchical mixture of experts [18–20]. Investigating soft tree models not only advances our understanding of this particular structure but also contributes to broader research into essential technological components critical for the development of large-scale language models [21].

A research question that we tackle in this paper is: "Can LMC be achieved for soft tree ensembles?". Our empirical results, which are highlighted with a green line in the top left panel of Figure 1, clearly show that the answer is "Yes". This plot shows the variation in test accuracy when interpolating weights of soft oblivious trees, perfect binary soft trees with shared parameters at each depth, trained from different random initializations. The green line is obtained by our method introduced in this paper, where there is almost zero performance degradation. Furthermore, as shown in the bottom left panel of Figure 1, the performance can even improve when interpolating between models trained on split datasets.

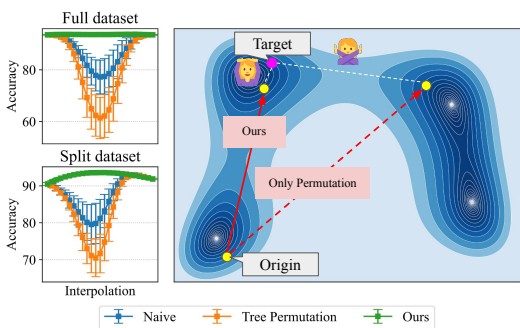

Figure 1: A representative experimental result on the MiniBooNE [22] dataset (left) and conceptual diagram of the LMC for tree ensembles (right).

The key insight is that, when performing interpolation between two model parameters, considering only tree permutation invariance, which corresponds to the permutation invariance of neural networks, is *not sufficient* to achieve LMC, as shown in the orange lines in the plots. An intuitive understanding of this situation is also illustrated in the right panel of Figure 1. To achieve LMC, that is, the green lines, we show that two additional invariances beyond tree permutation, *subtree flip invariance* and *splitting order invariance*, which inherently exist for tree architectures, should be accounted for.

Moreover, we demonstrate that it is possible to exclude such additional invariances while preserving LMC by modifying tree architectures. We realize such an architecture based on *a decision list*, a binary tree structure where branches extend in only one direction. By designating one of the terminal leaves as an empty node, we introduce a customized decision list that omits both subtree flip invariance and splitting order invariance, and empirically show that this can achieve LMC by considering only tree permutation invariance. Since incorporating additional invariances is computationally expensive, we can efficiently perform weight-space averaging in model merging on our customized decision lists.

Our contributions are summarized as follows:

- First achievement of LMC for tree ensembles with accounting for additional invariances beyond tree permutation.
- Development of a decision list-based tree architecture that does not involve the additional invariances.
- A thorough empirical investigation of LMC across various tree architectures, invariances, and real-world datasets.

## 2   Preliminary

We prepare the basic concepts of LMC and soft tree ensembles.

### 2.1   Linear Mode Connectivity

Let us consider two models, $A$ and $B$, that have the same architecture. In the context of evaluating LMC, the concept of a "barrier" is frequently used [4, 23]. Let $\Theta_A, \Theta_B \in \mathbb{R}^P$ be vectorized parameters of models $A$ and $B$, respectively, for $P$ parameters. Assume that $\mathcal{C} : \mathbb{R}^P \to \mathbb{R}$ measures the performance of the model, such as accuracy, given its parameter vector. If higher values of $\mathcal{C}(\cdot)$

mean better performance, the barrier between two parameter vectors $\boldsymbol{\Theta}_A$ and $\boldsymbol{\Theta}_B$ is defined as:

$$\mathcal{B}(\boldsymbol{\Theta}_A, \boldsymbol{\Theta}_B) = \sup_{\lambda \in [0,1]} \left[ \lambda \mathcal{C}(\boldsymbol{\Theta}_A) + (1 - \lambda)\mathcal{C}(\boldsymbol{\Theta}_B) - \mathcal{C}(\lambda \boldsymbol{\Theta}_A + (1 - \lambda)\boldsymbol{\Theta}_B) \right]. \tag{1}$$

We can simply reverse the subtraction order if lower values of $\mathcal{C}(\cdot)$ mean better performance like loss.

Several techniques have been developed to reduce barriers by transforming parameters while preserving functional equivalence. Two main approaches are *activation matching* (AM) and *weight matching* (WM). AM takes the behavior of model inference into account, while WM simply compares two models using their parameters. The validity of both AM and WM has been theoretically supported [24]. Numerous algorithms are available for implementing AM and WM. For instance, [4] uses a formulation based on the Linear Assignment Problem (LAP) to find suitable permutations, while [23] employs a differentiable formulation that allows for the optimization of permutations using gradient-based methods.

Existing research has focused exclusively on neural network architectures such as multi-layer perceptrons (MLP) and convolutional neural networks (CNN). No study has been conducted from the perspective of linear mode connectivity for soft tree ensembles.

## 2.2 Soft Tree Ensemble

Unlike typical hard decision trees, which explicitly determine the data flow to the right or left at each splitting node, soft trees represent the proportion of data flowing to the right or left as continuous values between 0 and 1. This approach enables a differentiable formulation.

We use a sigmoid function, $\sigma : \mathbb{R} \to (0,1)$ to formulate a function $\mu_{m,\ell}(\boldsymbol{x}_i, \boldsymbol{w}_m, \boldsymbol{b}_m) : \mathbb{R}^F \times \mathbb{R}^{F \times \mathcal{N}} \times \mathbb{R}^{1 \times \mathcal{N}} \to (0,1)$ that represents the proportion of the $i$th data point $\boldsymbol{x}_i$ flowing to the $\ell$th leaf of the $m$th tree as a result of soft splittings:

$$\mu_{m,\ell}(\boldsymbol{x}_i, \boldsymbol{w}_m, \boldsymbol{b}_m) = \prod_{n=1}^{\mathcal{N}} \underbrace{\sigma(\boldsymbol{w}_{m,n}^\top \boldsymbol{x}_i + b_{m,n})}_{\text{flow to the left}}^{\mathbb{1}_{\ell \swarrow n}} \underbrace{(1 - \sigma(\boldsymbol{w}_{m,n}^\top \boldsymbol{x}_i + b_{m,n}))}_{\text{flow to the right}}^{\mathbb{1}_{n \searrow \ell}}, \tag{2}$$

where $\mathcal{N}$ denotes the number of splitting nodes in each tree. The parameters $\boldsymbol{w}_{m,n} \in \mathbb{R}^F$ and $b_{m,n} \in \mathbb{R}$ correspond to the feature selection mask and splitting threshold value for $n$th node in a $m$th tree, respectively. The expression $\mathbb{1}_{\ell \swarrow n}$ (resp. $\mathbb{1}_{n \searrow \ell}$) is an indicator function that returns 1 if the $\ell$th leaf is positioned to the left (resp. right) of a node $n$, and 0 otherwise.

If parameters are shared across all splitting nodes at the same depth, such perfect binary trees are called *oblivious trees*. Mathematically, $\boldsymbol{w}_{m,n} = \boldsymbol{w}_{m,n'}$ and $b_{m,n} = b_{m,n'}$ for any nodes $n$ and $n'$ at the same depth in an oblivious tree. Oblivious trees can significantly reduce the number of parameters from an exponential to a linear order of the tree depth, and they are actively used in practice [9, 11].

To classify $C$ categories, the output of the $m$th tree is computed by the function $f_m : \mathbb{R}^F \times \mathbb{R}^{F \times \mathcal{N}} \times \mathbb{R}^{1 \times \mathcal{N}} \times \mathbb{R}^{C \times \mathcal{L}} \to \mathbb{R}^C$ as sum of the leaf parameters $\boldsymbol{\pi}_{m,\ell}$ weighted by the outputs of $\mu_{m,\ell}(\boldsymbol{x}_i, \boldsymbol{w}_m, \boldsymbol{b}_m)$:

$$f_m(\boldsymbol{x}_i, \boldsymbol{w}_m, \boldsymbol{b}_m, \boldsymbol{\pi}_m) = \sum_{\ell=1}^{\mathcal{L}} \boldsymbol{\pi}_{m,\ell} \mu_{m,\ell}(\boldsymbol{x}_i, \boldsymbol{w}_m, \boldsymbol{b}_m), \tag{3}$$

where $\mathcal{L}$ is the number of leaves in a tree. By combining this function for $M$ trees, we realize the function $f : \mathbb{R}^F \times \mathbb{R}^{M \times F \times \mathcal{N}} \times \mathbb{R}^{M \times 1 \times \mathcal{N}} \times \mathbb{R}^{M \times C \times \mathcal{L}} \to \mathbb{R}^C$ as an ensemble model consisting of $M$ trees:

$$f(\boldsymbol{x}_i, \boldsymbol{w}, \boldsymbol{b}, \boldsymbol{\pi}) = \sum_{m=1}^{M} f_m(\boldsymbol{x}_i, \boldsymbol{w}_m, \boldsymbol{b}_m, \boldsymbol{\pi}_m), \tag{4}$$

with the parameters $\boldsymbol{w} = (\boldsymbol{w}_1, \ldots, \boldsymbol{w}_M)$, $\boldsymbol{b} = (\boldsymbol{b}_1, \ldots, \boldsymbol{b}_M)$, and $\boldsymbol{\pi} = (\boldsymbol{\pi}_1, \ldots, \boldsymbol{\pi}_M)$ being randomly initialized.

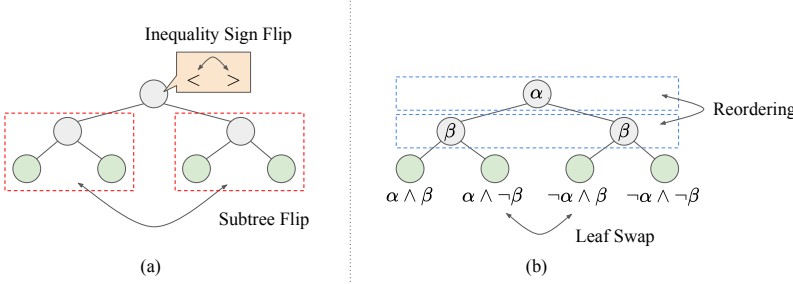

Figure 2: (a) Subtree flip invariance. (b) Splitting order invariance for an oblivious tree.

Despite the apparent differences, there are correspondences between MLPs and soft tree ensemble models. The formulation of a soft tree ensemble with $D = 1$ is:

$$f(\boldsymbol{x}_i, \boldsymbol{w}, \boldsymbol{b}, \boldsymbol{\pi}) = \sum_{m=1}^{M} \Big( \sigma(\boldsymbol{w}_{m,1}^{\top} \boldsymbol{x}_i + b_{m,1}) \boldsymbol{\pi}_{m,1} + (1 - \sigma(\boldsymbol{w}_{m,1}^{\top} \boldsymbol{x}_i + b_{m,1})) \boldsymbol{\pi}_{m,2} \Big)$$

$$= \sum_{m=1}^{M} \Big( (\boldsymbol{\pi}_{m,1} - \boldsymbol{\pi}_{m,2}) \sigma(\boldsymbol{w}_{m,1}^{\top} \boldsymbol{x}_i + b_{m,1}) + \boldsymbol{\pi}_{m,2} \Big). \tag{5}$$

When we consider the correspondence between $\boldsymbol{\pi}_{m,1} - \boldsymbol{\pi}_{m,2}$ in tree ensembles and second layer weights in the two-layer perceptron, the tree ensembles model matches to the two-layer perceptron. It is clear from the formulation that the permutation of hidden neurons in a neural network corresponds to the rearrangement of trees in a tree ensemble.

## 3  Invariances Inherent to Tree Ensembles

In this section, we discuss additional invariances inherent to trees (Section 3.1) and introduce a matching strategy specifically for tree ensembles (Section 3.2). We also show that the presence of additional invariances varies depending on the tree structure, and we present tree structures where no additional invariances beyond tree permutation exist (Section 3.3).

### 3.1  Parameter modification processes that maintains functional equivalence in tree ensembles

First, we clarify what invariances should be considered for tree ensembles, which are expected to reduce the barrier significantly if taken into account. When we consider perfect binary trees, there are three types of invariance:

- **Tree permutation invariance.** In Equation (4), the behavior of the function does not change even if the order of the $M$ trees is altered. This corresponds to the permutation of internal nodes in neural networks, which has been a subject of active interest in previous studies on LMC.

- **Subtree flip invariance.** When the left and right subtrees are swapped simultaneously with the inversion of the inequality sign at the split, the functional behavior remains unchanged, which we refer to as *subtree flip invariance*. Figure 2(a) presents a schematic diagram of this invariance, which is not found in neural networks but is unique to binary tree-based models. Since $\sigma(-c) = 1 - \sigma(c)$ for $c \in \mathbb{R}$ due to the symmetry of sigmoid, the inversion of the inequality is achieved by inverting the signs of $\boldsymbol{w}_{m,n}$ and $b_{m,n}$. [25] also focused on the sign of weights, but in a different way from ours. They pay attention to the amount of change from the parameters at the start of fine-tuning, rather than discussing the sign of the parameters.

- **Splitting order invariance.** Oblivious trees share parameters at the same depth, which means that the decision boundaries are straight lines without any bends. With this characteristic, even if the splitting rules at different depths are swapped, functional equivalence can be achieved if the positions of leaves are also swapped appropriately as shown in Figure 2(b). This invariance does not exist for non-oblivious perfect binary trees without parameter sharing, as the behavior of the decision boundary varies depending on the splitting order.

Note that MLPs also have an additional invariance beyond just permutation. Particularly in MLPs that employ ReLU as an activation function, the output of each layer changes linearly with a zero crossover. Therefore, it is possible to modify parameters without changing functional behavior by multiplying the weights in one layer by a constant and dividing the weights in the previous layer by the same constant. However, since the soft tree is based on the sigmoid function, this invariance does not apply. Previous studies [3, 4, 23] have consistently achieved significant reductions in barriers without accounting for this scale invariance. One potential reason is that changes in parameter scale are unlikely due to the nature of optimization via gradient descent. Conversely, when we consider additional invariances inherent to trees, the scale is equivalent to the original parameters.

## 3.2 Matching Strategy

Here, we propose a matching strategy for binary trees. When considering invariances, it is necessary to compare multiple functionally equivalent trees and select the most suitable one for achieving LMC. Although comparing tree parameters is a straightforward approach, since the contribution of all the parameters in a tree is not equal, we apply weighting for each node for better matching. By interpreting a tree as a rule set with shared parameters as shown in Figure 3, we determine the weight of each splitting node

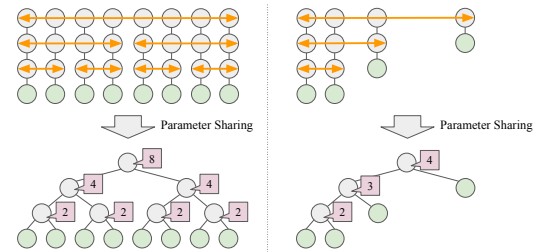

Figure 3: Weighting strategy.

by counting the number of leaves to which the node affects. For example, in the case of the left example in Figure 3, the root node affects eight leaves, nodes at depth 2 affect four leaves, and nodes at depth 3 affect two leaves. This strategy can apply to even trees other than perfect binary trees. For example, in the right example of Figure 3, the root node affects four leaves, a node at depth 2 affects three leaves, and a node at depth 3 affects two leaves.

In this paper, we employ the LAP, which is used as a standard benchmark [4] for matching algorithms. The procedures for AM and WM are as follows. Detailed algorithms (Algorithms 1 and 2) are described in Section A in the supplementary material.

- **Activation Matching (Algorithm 1).** In trees, there is nothing that directly corresponds to the activations in neural networks. However, by treating the output of each individual tree as an activation value of a neural network, it is possible to optimize the permutation of trees while examining their output similarities. Regarding subtree flip and splitting order invariances, it is possible to find the optimal pattern from all the possible patterns of flips and changes in the splitting order. Since the tree-wise output remains unchanged, the similarity between each tree, generated by considering additional invariances, and the target tree is evaluated based on the inner product of parameters while applying node-wise weighting.

- **Weight Matching (Algorithm 2).** Similar to AM, WM also involves applying weighting while extracting the optimal pattern by exploring possible flipping and ordering patterns. Although it is necessary to solve the LAP multiple times for each layer in MLPs [4], tree ensembles require only a single run of the LAP since there are no layers.

The time complexity of solving the LAP is $\mathcal{O}(M^3)$ using a modified Jonker-Volgenant algorithm without initialization [26], implemented in SciPy [27], where $M$ is the number of trees. If only considering tree permutation, this process needs to be performed only once in both WM and AM. However, when considering additional invariances, we need to solve the LAP for each pattern generated by considering these additional invariances. In a non-oblivious perfect binary tree with depth $D$, there are $2^D - 1$ splitting nodes, leading to $2^{2^D-1}$ possible combinations of sign flips. Additionally, in the case of oblivious trees, there are $D!$ different patterns of splitting order invariance. Therefore, for large values of $D$, conducting a brute-force search becomes impractical.

In Section 3.3, we will discuss methods to eliminate additional invariance by adjusting the tree structure. This enables efficient matching even for deep models. Additionally, in Section 4.2, we will present numerical experiment results and discuss that the practical motivation to apply these algorithms is limited when targeting deep perfect binary trees.

### 3.3 Architecture-dependency of the Invariances

In previous subsections, tree architectures are fixed to perfect binary trees as they are most commonly and practically used in soft trees. However, tree architectures can be flexible as we have shown in the right example in Figure 3, and here we show that we can specifically design tree architecture that has neither the subtree flip nor splitting order invariances. This allows efficient matching as considering such two invariances is computationally expensive.

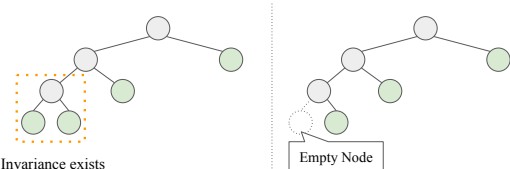

Figure 4: Tree architecture where neither subtree flip invariance nor splitting order invariance exists.

Our idea is to modify a *decision list* shown on the left side of Figure 4, which is a tree structure where branches extend in only one direction. Due to this asymmetric structure, the number of parameters does not increase exponentially with the depth, and the splitting order invariance does not exist. Moreover, subtree flip invariance also does not exist for any internal nodes except for the terminal splitting node, as shown in the left side of Figure 4. To completely remove this invariance, we virtually eliminate one of the terminal leaves by leaving the node empty, that is, a fixed prediction value of zero, as shown on the right side of Figure 4. Therefore only permutation invariance exists for our proposed architecture. We summarize invariances inherent to each model architecture in Table 1.

Table 1: Invariances inherent to each model architecture.

|  | Perm | Flip | Order |
|---|---|---|---|
| Non-Oblivious Tree | ✓ | ✓ | ✗ |
| Oblivious Tree | ✓ | ✓ | ✓ |
| Decision List | ✓ | (✓) | ✗ |
| Decision List (Modified) | ✓ | ✗ | ✗ |

## 4 Experiment

We empirically evaluate barriers in soft tree ensembles to examine LMC.

### 4.1 Setup

**Datasets.** In our experiments, we employed Tabular-Benchmark [28], a collection of tabular datasets suitable for evaluating tree ensembles. Details of datasets are provided in Section B in the supplementary material. As proposed in [28], we randomly sampled $10,000$ instances for train and test data from each dataset. If the dataset contains fewer than $20,000$ instances, they are randomly divided into halves for train and test data. We applied quantile transformation to each feature and standardized it to follow a normal distribution.

**Hyperparameters.** We used three different learning rates $\eta \in \{0.01, 0.001, 0.0001\}$ and adopted the one that yields the highest training accuracy for each dataset. The batch size is set at $512$. It is known that the optimal settings for the learning rate and batch size are interdependent [29]. Therefore, it is reasonable to fix the batch size while adjusting the learning rate. During AM, we set the amount of data used for random sampling to be the same as the batch size, thus using $512$ samples to measure the similarity of the tree outputs. As the number of trees $M$ and their depths $D$ vary for each experiment, these details will be specified in the experimental results section. During training, we minimized cross-entropy using Adam [16] with its default hyperparameters[1]. Training is conducted for $50$ epochs. To measure the barrier using Equation (1), experiments were conducted by interpolating between two models with $\lambda \in \{0, 1/24, \ldots, 23/24, 1\}$, which has the same granularity as in [4].

**Randomness.** We conducted experiments with five different random seed pairs: $r_A \in \{1, 3, 5, 7, 9\}$ and $r_B \in \{2, 4, 6, 8, 10\}$. As a result, the initial parameters and the contents of the data mini-batches during training are different in each training. In contrast to spawning [5] that branches off from the exact same model partway through, we used more challenging practical conditions. The parameters $w$, $b$, and $\pi$ were randomly initialized using a uniform distribution, identical to the procedure for a fully connected layer in the MLP[2].

---

[1] https://pytorch.org/docs/stable/generated/torch.optim.Adam.html
[2] https://pytorch.org/docs/stable/generated/torch.nn.Linear.html

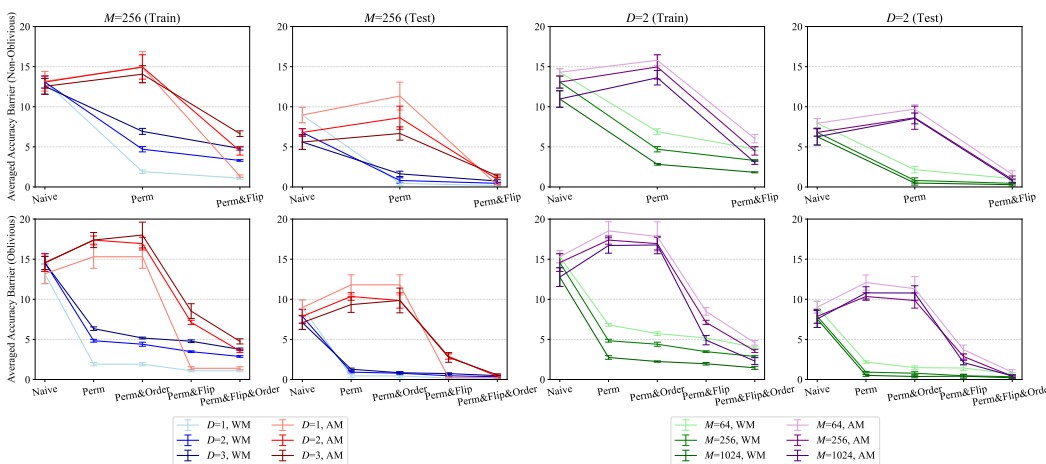

Figure 5: Barriers averaged across 16 datasets with respect to considered invariances for non-oblivious (top row) and oblivious (bottom row) trees. The error bars show the standard deviations of 5 executions.

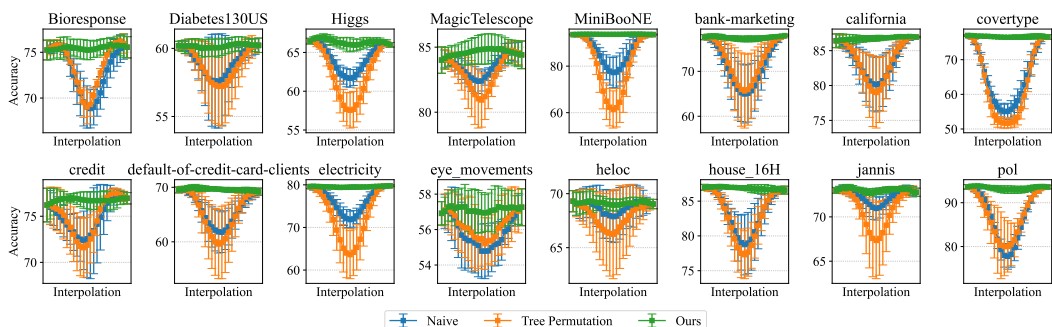

Figure 6: Interpolation curves of test accuracy for oblivious trees on 16 datasets from Tabular-Benchmark [28]. Two model pairs are trained with on the same dataset. The error bars show the standard deviations of 5 executions. We used $M = 256$ trees with a depth $D = 2$.

**Resources.** All experiments were conducted on a system equipped with an Intel Xeon E5-2698 CPU at 2.20 GHz, 252 GB of memory, and Tesla V100-DGXS-32GB GPU, running Ubuntu Linux (version 4.15.0-117-generic). The reproducible PyTorch [30] implementation is provided in the supplementary material.

## 4.2 Results for Perfect Binary Trees

Figure 5 shows how the barrier between two perfect binary tree model pairs changes in each operation. The vertical axis of each plot in Figure 5 shows the averaged barrier over datasets for each considered invariance. The results for both the oblivious and non-oblivious trees are plotted separately in a vertical layout. The panels on the left display the results when the depth $D$ of the tree varies, keeping $M = 256$ constant. The panels on the right show the results when the number of trees $M$ varies, with $D$ fixed at 2. For both oblivious and non-oblivious trees, we observed that the barrier significantly decreases as the considered invariances increase. Focusing on the test data results, after accounting for various invariances, the barrier is nearly zero, indicating that LMC has been achieved. In particular, the difference between the case of only permutation and the case where additional invariances are considered tends to be larger in the case of AM. This is because parameter values are not used during the rearrangement of the tree in AM. Additionally, it has been observed that the barrier increases as trees become deeper, and the barrier decreases as the number of trees increases. These behaviors correspond to the changes observed in neural networks when the depth varies or when the width of hidden layers increases [3, 4]. Figure 6 shows interpolation curves when using AM in oblivious trees

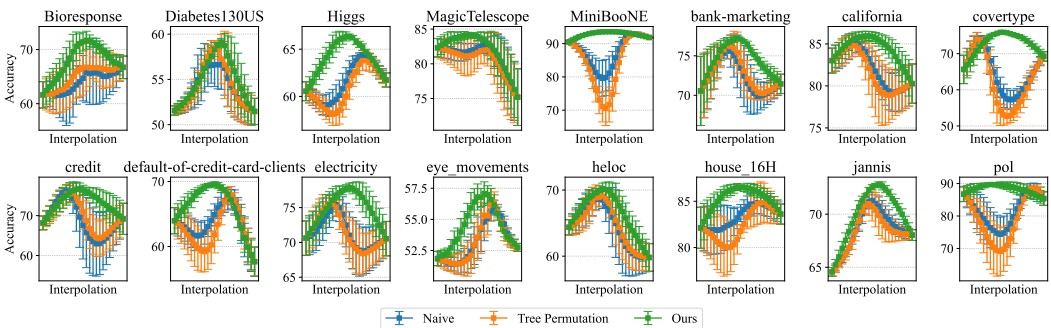

Figure 7: Interpolation curves of test accuracy for oblivious trees on 16 datasets from Tabular-Benchmark [28]. Two model pairs are trained on split datasets with different class ratios. The error bars show the standard deviations of 5 executions. We used $M = 256$ trees with a depth $D = 2$.

with $D = 2$ and $M = 256$. Other detailed results, such as performance for each dataset, are provided in Section C in the supplementary material.

Furthermore, we conducted experiments with split data following the protocol in [4, 31], where the initial split consists of randomly sampled 80% negative and 20% positive instances, and the second split inverts these ratios. There is no overlap between the two split datasets. We trained two model pairs using these separately split datasets and observed an improvement in performance by interpolating their parameters. Figure 7 illustrates the interpolation curves under AM in oblivious trees with parameters $D = 2$ and $M = 256$. We can observe that considering additional invariances improves performance after interpolation. Note that the data split is configured to remain consistent even when the training random seeds differ. Detailed results for each dataset using WM or AM are provided in Section C of the supplementary material.

Table 2 compares the average test barriers of an MLP with a ReLU activation function, whose width is equal to the number of trees, $M = 256$. The procedure for MLPs follows that described in Section 4.1. The permutation for MLPs is optimized using the method described in [4]. Since [4] indicated that WM outperforms AM in neural networks, WM was used for the comparison. Overall, tree models exhibit smaller barriers compared to MLPs while keeping similar accuracy levels. It is important to note that MLPs with $D > 1$ tend to have more parameters at the same depth compared to trees, leading to more complex optimization landscapes. Nevertheless, the barrier for the non-oblivious tree at $D = 3$ is smaller than that for the MLP at $D = 2$, even with more parameters. Furthermore, at the same depth of $D = 1$, tree models have a smaller barrier. Here, the model size is evaluated using $F = 44$, the average input feature size of 16 datasets used in the experiments.

Table 2: Barriers, accuracies, and model sizes for MLP, non-oblivious trees, and oblivious trees.

| | **MLP** | | | |
|---|---|---|---|---|
| **Depth** | **Barrier** | | **Accuracy** | **Size** |
| | **Naive** | **Perm** [4] | | |
| 1 | $8.755 \pm 0.877$ | $\underline{0.491 \pm 0.062}$ | $76.286 \pm 0.094$ | 12034 |
| 2 | $15.341 \pm 1.125$ | $\underline{2.997 \pm 0.709}$ | $75.981 \pm 0.139$ | 77826 |
| 3 | $15.915 \pm 2.479$ | $\underline{5.940 \pm 2.153}$ | $75.935 \pm 0.117$ | 143618 |

| | **Non-Oblivious Tree** | | | | |
|---|---|---|---|---|---|
| **Depth** | **Barrier** | | | **Accuracy** | **Size** |
| | **Naive** | **Perm** | **Ours** | | |
| 1 | $8.965 \pm 0.963$ | $0.449 \pm 0.235$ | $\underline{0.181 \pm 0.078}$ | $76.464 \pm 0.167$ | 12544 |
| 2 | $6.801 \pm 0.464$ | $0.811 \pm 0.333$ | $\underline{0.455 \pm 0.105}$ | $76.631 \pm 0.052$ | 36608 |
| 3 | $5.602 \pm 0.926$ | $1.635 \pm 0.334$ | $\underline{0.740 \pm 0.158}$ | $76.339 \pm 0.115$ | 84736 |

| | **Oblivious Tree** | | | | |
|---|---|---|---|---|---|
| **Depth** | **Barrier** | | | **Accuracy** | **Size** |
| | **Naive** | **Perm** | **Ours** | | |
| 1 | $8.965 \pm 0.963$ | $0.449 \pm 0.235$ | $\underline{0.181 \pm 0.078}$ | $76.464 \pm 0.167$ | 12544 |
| 2 | $7.881 \pm 0.866$ | $0.918 \pm 0.092$ | $\underline{0.348 \pm 0.172}$ | $76.623 \pm 0.042$ | 25088 |
| 3 | $7.096 \pm 0.856$ | $1.283 \pm 0.139$ | $\underline{0.484 \pm 0.049}$ | $76.535 \pm 0.063$ | 38656 |

In Section 3.2, we have shown that considering additional invariances for deep perfect binary trees is computationally challenging, which may suggest developing heuristic algorithms for deep trees. However, we consider it is rather a low priority, supported by our observations that the barrier tends to increase as trees deepen even if we consider invariances. This trend indicates that deep models are fundamentally less important for model merging considerations. Furthermore, deep perfect binary trees are rarely used in practical scenarios. [12] have demonstrated that generalization performance degrades with increasing depth in perfect binary trees due to the degeneracy of the Neural Tangent Kernel (NTK) [32]. This evidence further supports the preference for shallow perfect binary trees, and increasing the number of trees can enhance the expressive power while reducing barriers.

Table 3: Barriers averaged for 16 datasets under WM with $D = 2$ and $M = 256$.

| Architecture | Train | | | | Test | | | |
|---|---|---|---|---|---|---|---|---|
| | Barrier | | | Accuracy | Barrier | | | Accuracy |
| | Naive | Perm | Ours | | Naive | Perm | Ours | |
| Non-Oblivious Tree | 13.079 ± 0.755 | 4.707 ± 0.332 | 3.303 ± 0.104 | 85.646 ± 0.090 | 6.801 ± 0.464 | 0.811 ± 0.333 | 0.455 ± 0.105 | 76.631 ± 0.052 |
| Oblivious Tree | 14.580 ± 1.108 | 4.834 ± 0.176 | 2.874 ± 0.108 | 85.808 ± 0.146 | 7.881 ± 0.866 | 0.919 ± 0.093 | 0.348 ± 0.172 | 76.623 ± 0.042 |
| Decision List | 13.835 ± 0.788 | 3.687 ± 0.230 | — | 85.337 ± 0.134 | 7.513 ± 0.944 | 0.436 ± 0.120 | — | 76.629 ± 0.119 |
| Decision List (Modified) | 12.922 ± 1.131 | 3.328 ± 0.204 | — | 85.563 ± 0.141 | 6.734 ± 1.096 | 0.468 ± 0.150 | — | 76.773 ± 0.051 |

Table 4: Barriers averaged for 16 datasets under AM with $D = 2$ and $M = 256$.

| Architecture | Train | | | | Test | | | |
|---|---|---|---|---|---|---|---|---|
| | Barrier | | | Accuracy | Barrier | | | Accuracy |
| | Naive | Perm | Ours | | Naive | Perm | Ours | |
| Non-Oblivious Tree | 13.079 ± 0.755 | 14.963 ± 1.520 | 4.500 ± 0.527 | 85.646 ± 0.090 | 6.801 ± 0.464 | 8.631 ± 1.444 | 0.943 ± 0.435 | 76.631 ± 0.052 |
| Oblivious Tree | 14.580 ± 1.108 | 17.380 ± 0.509 | 3.557 ± 0.201 | 85.808 ± 0.146 | 7.881 ± 0.866 | 10.349 ± 0.476 | 0.395 ± 0.185 | 76.623 ± 0.042 |
| Decision List | 13.835 ± 0.788 | 12.785 ± 1.924 | — | 85.337 ± 0.134 | 7.513 ± 0.944 | 7.452 ± 1.840 | — | 76.629 ± 0.119 |
| Decision List (Modified) | 12.922 ± 1.131 | 6.364 ± 0.194 | — | 85.563 ± 0.141 | 6.734 ± 1.096 | 2.114 ± 0.243 | — | 76.773 ± 0.051 |

### 4.3 Results for Decision Lists

We present empirical results of the original decision lists and our modified decision lists, as shown in Figure 4. As we have shown in Table 1, they have fewer invariances.

Figure 8 illustrates barriers as a function of depth, considering only permutation invariance, with $M$ fixed at 256. In this experiment, we have excluded non-oblivious trees from comparison as the number of their parameters exponentially increases as trees deepen, making them infeasible computation. Our proposed modified decision lists reduce the barrier more effectively than both oblivious trees and the original decision lists. However, barriers of the modified decision lists are still larger than those obtained by considering additional invariances with perfect binary trees. Tables 3 and 4 show the averaged barriers for 16 datasets, with $D = 2$ and $M = 256$. Although barriers of modified decision lists are small when considering only permutations (Perm), perfect binary trees such as oblivious trees with additional invariances (Ours) exhibit smaller barriers, which supports the validity of using oblivious trees as in [9, 11]. To summarize, when considering the practical use of model merging, if the goal is to prioritize efficient computation, we recommend using our proposed decision list. Conversely, if the goal is to prioritize barriers, it would be preferable to use perfect binary trees, which have a greater number of invariant operations that maintain the functional behavior.

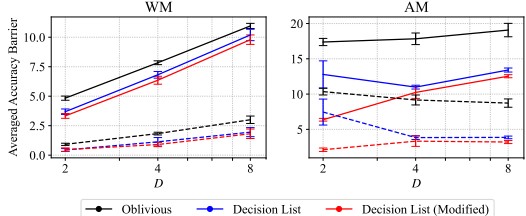

Figure 8: Averaged barrier for 16 datasets as a function of tree depth. The error bars show the standard deviations of 5 executions. The solid line represents the barrier in train accuracy, while the dashed line represents the barrier in test accuracy.

## 5 Conclusion

We have presented the first investigation of LMC for soft tree ensembles. We have identified additional invariances inherent in tree architectures and empirically demonstrated the importance of considering these factors. Achieving LMC is crucial not only for understanding the behavior of non-convex optimization from a learning theory perspective but also for implementing practical techniques such as model merging. By arithmetically combining parameters of differently trained models, a wide range of applications such as task-arithmetic [33], including unlearning [34] and continual-learning [35], have been explored. Our research extends these techniques to soft tree ensembles that began training from entirely different initial conditions. We will leave these empirical investigations for future work.

This study provides a fundamental analysis of ensemble learning, and we believe that our discussion will not have any negative societal impacts.

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

# A  Detailed Algorithms

We present pseudo-code of algorithms for activation matching (Algorithm 1) and weight matching (Algorithm 2). In these algorithms, if there is only one possible pattern for $U \in \mathbb{N}$, which represents the number of possible operations, and the corresponding operation does nothing in particular, it becomes equivalent to simply considering tree permutations.

---

**Algorithm 1:** Activation matching for soft trees

---

1   ACTIVATIONMATCHING($\boldsymbol{\Theta}_A \in \mathbb{R}^{M \times P_{\text{Tree}}}$, $\boldsymbol{\Theta}_B \in \mathbb{R}^{M \times P_{\text{Tree}}}$, $\boldsymbol{x}_{\text{sampled}} \in \mathbb{R}^{F \times N_{\text{sampled}}}$)

2      Initialize $\boldsymbol{O}_A \in \mathbb{R}^{M \times N_{\text{sampled}} \times C}$ and $\boldsymbol{O}_B \in \mathbb{R}^{M \times N_{\text{sampled}} \times C}$ to store outputs

3      **for** $m = 1$ *to* $M$ **do**

4          **for** $i = 1$ *to* $N_{\text{sampled}}$ **do**

5              Set the output of the $m$th tree with $\boldsymbol{\Theta}_A[m]$ using $\boldsymbol{x}_{\text{sampled}}[:, i]$ to $\boldsymbol{O}_A[m, i]$.

6              Set the output of the $m$th tree with $\boldsymbol{\Theta}_B[m]$ using $\boldsymbol{x}_{\text{sampled}}[:, i]$ to $\boldsymbol{O}_B[m, i]$.

7      Initialize similarity matrix $\boldsymbol{S} \in \mathbb{R}^{M \times M}$

8      **for** $m_A = 1$ *to* $M$ **do**

9          **for** $m_B = 1$ *to* $M$ **do**

10              $\boldsymbol{S}[m_A, m_B] \leftarrow$ FLATTEN($\boldsymbol{O}_A[m_A]$) $\cdot$ FLATTEN($\boldsymbol{O}_B[m_B]$)

11      $\boldsymbol{p} \leftarrow$ LINEARSUMASSIGNMENT($\boldsymbol{S}$)          // $\boldsymbol{p} \in \mathbb{N}^M$: Optimal assignments

12      $\boldsymbol{\Theta}_A, \boldsymbol{\Theta}_B \leftarrow$ WEIGHTING($\boldsymbol{\Theta}_A, \boldsymbol{\Theta}_B$)

13      Initialize operation indices $\boldsymbol{q} \in \mathbb{N}^M$

14      **for** $m = 1$ *to* $M$ **do**

15          **for** $u = 1$ *to* $U$ **do**          // $U \in \mathbb{N}$: Number of possible operations

16              $u' \leftarrow$ UPDATEBESTOPERATION(ADJUSTTREE($\boldsymbol{\Theta}_A[m], u$) $\cdot \boldsymbol{\Theta}_B[m], u$)

17          Append $u'$ to $\boldsymbol{q}$          // $\boldsymbol{q} \in \mathbb{N}^M$: Optimal operations

18      **return** $\boldsymbol{p}, \boldsymbol{q}$

---

 

---

**Algorithm 2:** Weight matching for soft trees

---

1   WEIGHTMATCHING($\boldsymbol{\Theta}_A \in \mathbb{R}^{M \times P_{\text{Tree}}}$, $\boldsymbol{\Theta}_B \in \mathbb{R}^{M \times P_{\text{Tree}}}$)

2      $\boldsymbol{\Theta}_A, \boldsymbol{\Theta}_B \leftarrow$ WEIGHTING($\boldsymbol{\Theta}_A, \boldsymbol{\Theta}_B$)

3      Initialize similarity matrix for each operation $\boldsymbol{S} \in \mathbb{R}^{U \times M \times M}$

4      **for** $u = 1$ *to* $U$ **do**

5          **for** $m_A = 1$ *to* $M$ **do**

6              $\boldsymbol{\theta} \leftarrow$ ADJUSTTREE($\boldsymbol{\Theta}_A[m_A], u$)     // $\boldsymbol{\theta} \in \mathbb{R}^{P_{\text{Tree}}}$: Adjusted tree-wise parameters

7              **for** $m_B = 1$ *to* $M$ **do**

8                  $\boldsymbol{S}[u, m_A, m_B] \leftarrow \boldsymbol{\theta} \cdot \boldsymbol{\Theta}_B[m_B]$

9      $\boldsymbol{S}' \leftarrow \max(\boldsymbol{S}, \text{axis=0})$          // $\boldsymbol{S}' \in \mathbb{R}^{M \times M}$: Similarity matrix between trees

10      $\boldsymbol{p} \leftarrow$ LINEARSUMASSIGNMENT($\boldsymbol{S}'$)          // $\boldsymbol{p} \in \mathbb{N}^M$: Optimal assignments

11      $\boldsymbol{q} \leftarrow \text{argmax}(\boldsymbol{S}, \text{axis=0})[\boldsymbol{p}]$          // $\boldsymbol{q} \in \mathbb{N}^M$: Optimal operations

12      **return** $\boldsymbol{p}, \boldsymbol{q}$

---

Here, we describe the specifications of the notations and functions used in Algorithms 1 and 2. In Section 2.1, $\boldsymbol{\Theta}_A$ and $\boldsymbol{\Theta}_B$ are initially defined as vectors. However, for ease of use, in Algorithms 1 and 2, $\boldsymbol{\Theta}_A$ and $\boldsymbol{\Theta}_B$ are represented as matrices of size $\mathbb{R}^{M \times P_{\text{Tree}}}$, where $P_{\text{Tree}}$ denotes the number of parameters in a single tree. Multidimensional array elements are accessed using square brackets $[\cdot]$. For example, for $\boldsymbol{G} \in \mathbb{R}^{I \times J}$, $\boldsymbol{G}[i]$ refers to the $i$th slice along the first dimension, and $\boldsymbol{G}[:, j]$ refers to the $j$th slice along the second dimension, with sizes $\mathbb{R}^J$ and $\mathbb{R}^I$, respectively. Furthermore, it can also accept a vector $\boldsymbol{v} \in \mathbb{N}^l$ as an input. In this case, $\boldsymbol{G}[\boldsymbol{v}] \in \mathbb{R}^{l \times J}$. The FLATTEN function converts multidimensional input into a one-dimensional vector format. As the LINEARSUMASSIGNMENT

function, scipy.optimize.linear_sum_assignment[3] is used to solve the LAP. In the ADJUSTTREE function, the parameters of a tree are modified according to the $u$th pattern among the enumerated $U$ patterns. Additionally, in the WEIGHTING function, parameters are multiplied by the square root of their weights defined in Section 3.2 to simulate the process of assessing a rule set. If the first argument for the UPDATEBESTOPERATION function, the input inner product, is larger than any previously input inner product values, then $u'$ is updated with $u$, the second argument. If not, $u'$ remains unchanged.

# B  Dataset

Table 5: Summary of the datasets used in the experiments.

| Dataset | $N$ | $F$ | Link |
|---|---|---|---|
| Bioresponse | 3434 | 419 | https://www.openml.org/d/45019 |
| Diabetes130US | 71090 | 7 | https://www.openml.org/d/45022 |
| Higgs | 940160 | 24 | https://www.openml.org/d/44129 |
| MagicTelescope | 13376 | 10 | https://www.openml.org/d/44125 |
| MiniBooNE | 72998 | 50 | https://www.openml.org/d/44128 |
| bank-marketing | 10578 | 7 | https://www.openml.org/d/44126 |
| california | 20634 | 8 | https://www.openml.org/d/45028 |
| covertype | 566602 | 10 | https://www.openml.org/d/44121 |
| credit | 16714 | 10 | https://www.openml.org/d/44089 |
| default-of-credit-card-clients | 13272 | 20 | https://www.openml.org/d/45020 |
| electricity | 38474 | 7 | https://www.openml.org/d/44120 |
| eye_movements | 7608 | 20 | https://www.openml.org/d/44130 |
| heloc | 10000 | 22 | https://www.openml.org/d/45026 |
| house_16H | 13488 | 16 | https://www.openml.org/d/44123 |
| jannis | 57580 | 54 | https://www.openml.org/d/45021 |
| pol | 10082 | 26 | https://www.openml.org/d/44122 |

# C  Additional Empirical Results

Tables 6, 7, 8 and 9 present the barrier for each dataset with $D = 2$ and $M = 256$. By incorporating additional invariances, it has been possible to consistently reduce the barriers.

Tables 10 and 11 detail the characteristics of the barriers in the decision lists for each dataset with $D = 2$ and $M = 256$. The barriers in the modified decision lists tend to be smaller.

Tables 12 and 13 show the barrier for each model when only considering permutations with $D = 2$ and $M = 256$. It is evident that focusing solely on permutations leads to smaller barriers in the modified decision lists compared to other architectures.

Figures 9, 10, 11, 12, 13, 14, 15 and 16 show the interpolation curves of oblivious trees with $D = 2$ and $M = 256$ across various datasets and configurations. Significant improvements are particularly noticeable in AM, but improvements are also observed in WM. These characteristics are also apparent in the non-oblivious trees, as shown in Figures 17, 18, 19, 20, 21, 22, 23 and 24. Regarding split data training, the dataset for each of the two classes is initially complete (100%). It is then divided into splits of 80% and 20%, and 20% and 80%, respectively. Each model is trained using these splits. Figures 13, 15, 21, and 23 show the training accuracy evaluated using the full dataset (100% for each class).

---

[3]https://docs.scipy.org/doc/scipy/reference/generated/scipy.optimize.linear_sum_assignment.html

Table 6: Accuracy barrier for non-oblivious trees with WM.

| Dataset | Train | | | Test | | |
|---|---|---|---|---|---|---|
| | Naive | Perm | Perm&Flip | Naive | Perm | Perm&Flip |
| Bioresponse | 18.944 ± 10.076 | 5.876 ± 1.477 | 4.132 ± 0.893 | 8.235 ± 6.456 | 1.285 ± 0.635 | 0.314 ± 0.432 |
| Diabetes130US | 2.148 ± 0.601 | 1.388 ± 1.159 | 0.947 ± 0.888 | 1.014 ± 0.959 | 0.540 ± 0.999 | 0.784 ± 0.840 |
| Higgs | 27.578 ± 1.742 | 18.470 ± 0.769 | 14.772 ± 1.419 | 4.055 ± 1.089 | 0.662 ± 0.590 | 0.292 ± 0.421 |
| MagicTelescope | 2.995 ± 1.198 | 0.576 ± 0.556 | 0.307 ± 0.346 | 2.096 ± 1.055 | 0.361 ± 0.618 | 0.229 ± 0.348 |
| MiniBooNE | 18.238 ± 4.570 | 2.272 ± 0.215 | 1.506 ± 0.211 | 12.592 ± 4.190 | 0.231 ± 0.314 | 0.000 ± 0.000 |
| bank-marketing | 13.999 ± 4.110 | 2.711 ± 1.183 | 1.521 ± 0.463 | 13.593 ± 4.567 | 1.843 ± 1.001 | 0.953 ± 0.688 |
| california | 6.396 ± 2.472 | 0.873 ± 0.551 | 0.520 ± 0.327 | 5.226 ± 2.377 | 0.224 ± 0.248 | 0.206 ± 0.131 |
| covertype | 16.823 ± 4.159 | 1.839 ± 0.336 | 0.914 ± 0.546 | 14.900 ± 4.016 | 1.035 ± 0.106 | 0.376 ± 0.333 |
| credit | 7.317 ± 2.425 | 3.172 ± 2.636 | 2.615 ± 0.831 | 5.861 ± 2.064 | 2.202 ± 3.103 | 1.830 ± 0.588 |
| default-of-credit-card-clients | 14.318 ± 4.509 | 5.419 ± 1.318 | 3.273 ± 0.793 | 6.227 ± 4.205 | 0.937 ± 1.036 | 0.243 ± 0.172 |
| electricity | 10.090 ± 2.930 | 1.035 ± 0.543 | 0.221 ± 0.192 | 9.422 ± 2.795 | 0.771 ± 0.478 | 0.130 ± 0.071 |
| eye_movements | 18.743 ± 1.994 | 11.605 ± 1.927 | 7.866 ± 1.301 | 1.495 ± 0.467 | 0.463 ± 0.183 | 0.180 ± 0.206 |
| heloc | 4.434 ± 1.611 | 1.652 ± 0.475 | 1.012 ± 0.481 | 0.830 ± 0.727 | 0.475 ± 0.447 | 0.322 ± 0.338 |
| house_16H | 8.935 ± 2.504 | 3.362 ± 0.482 | 2.660 ± 1.208 | 4.230 ± 2.189 | 0.219 ± 0.224 | 0.404 ± 0.782 |
| jannis | 17.756 ± 3.322 | 10.442 ± 1.404 | 7.362 ± 0.219 | 3.205 ± 2.849 | 0.029 ± 0.064 | 0.007 ± 0.016 |
| pol | 20.542 ± 2.873 | 4.612 ± 0.912 | 3.225 ± 1.080 | 15.830 ± 2.562 | 1.708 ± 0.599 | 1.012 ± 0.859 |

Table 7: Accuracy barrier for non-oblivious trees with AM.

| Dataset | Train | | | Test | | |
|---|---|---|---|---|---|---|
| | Naive | Perm | Perm&Flip | Naive | Perm | Perm&Flip |
| Bioresponse | 18.944 ± 10.076 | 14.066 ± 7.045 | 5.710 ± 0.915 | 8.235 ± 6.456 | 5.037 ± 3.141 | 0.966 ± 0.316 |
| Diabetes130US | 2.148 ± 0.601 | 3.086 ± 2.566 | 0.574 ± 0.365 | 1.014 ± 0.959 | 1.936 ± 2.878 | 0.105 ± 0.152 |
| Higgs | 27.578 ± 1.742 | 30.704 ± 2.899 | 18.435 ± 1.599 | 4.055 ± 1.089 | 7.272 ± 1.089 | 1.044 ± 0.483 |
| MagicTelescope | 2.995 ± 1.198 | 3.309 ± 1.486 | 0.778 ± 0.515 | 2.096 ± 1.055 | 2.693 ± 1.190 | 0.428 ± 0.327 |
| MiniBooNE | 18.238 ± 4.570 | 34.934 ± 8.157 | 2.332 ± 0.383 | 12.592 ± 4.190 | 28.721 ± 7.869 | 0.074 ± 0.081 |
| bank-marketing | 13.999 ± 4.110 | 13.598 ± 7.638 | 3.098 ± 0.539 | 13.593 ± 4.567 | 12.810 ± 7.605 | 2.643 ± 0.704 |
| california | 6.396 ± 2.472 | 5.800 ± 2.036 | 0.697 ± 0.535 | 5.226 ± 2.377 | 4.858 ± 2.017 | 0.261 ± 0.285 |
| covertype | 16.823 ± 4.159 | 19.708 ± 6.392 | 1.420 ± 0.619 | 14.900 ± 4.016 | 17.765 ± 6.400 | 0.758 ± 0.540 |
| credit | 7.317 ± 2.425 | 10.556 ± 8.753 | 3.640 ± 1.624 | 5.861 ± 2.064 | 9.378 ± 9.083 | 2.551 ± 1.987 |
| default-of-credit-card-clients | 14.318 ± 4.509 | 14.166 ± 2.297 | 4.247 ± 1.678 | 6.227 ± 4.205 | 6.514 ± 2.049 | 0.885 ± 1.852 |
| electricity | 10.090 ± 2.930 | 12.955 ± 4.558 | 0.762 ± 0.332 | 9.422 ± 2.795 | 12.261 ± 4.554 | 0.499 ± 0.260 |
| eye_movements | 18.743 ± 1.994 | 18.757 ± 1.273 | 10.957 ± 1.019 | 1.495 ± 0.467 | 1.583 ± 1.011 | 0.146 ± 0.167 |
| heloc | 4.434 ± 1.611 | 6.564 ± 2.404 | 1.774 ± 0.672 | 0.830 ± 0.727 | 2.179 ± 2.100 | 0.385 ± 0.370 |
| house_16H | 8.935 ± 2.504 | 10.184 ± 2.667 | 3.908 ± 0.863 | 4.230 ± 2.189 | 5.664 ± 2.461 | 1.056 ± 0.693 |
| jannis | 17.756 ± 3.322 | 19.004 ± 1.246 | 9.890 ± 1.036 | 3.205 ± 2.849 | 4.047 ± 1.415 | 0.346 ± 0.443 |
| pol | 20.542 ± 2.873 | 16.267 ± 3.914 | 7.967 ± 3.208 | 15.830 ± 2.562 | 12.863 ± 3.983 | 4.539 ± 2.727 |

Table 8: Accuracy barrier for oblivious trees with WM.

| Dataset | Train | | | Test | | |
|---|---|---|---|---|---|---|
| | Naive | Perm | Perm&Order&Flip | Naive | Perm | Perm&Order&Flip |
| Bioresponse | 16.642 ± 4.362 | 4.800 ± 0.895 | 3.289 ± 0.680 | 7.165 ± 2.547 | 1.069 ± 1.020 | 0.299 ± 0.247 |
| Diabetes130US | 3.170 ± 3.304 | 1.120 ± 1.123 | 0.246 ± 0.177 | 2.831 ± 3.476 | 0.882 ± 1.309 | 0.181 ± 0.155 |
| Higgs | 28.640 ± 0.914 | 19.754 ± 1.023 | 13.689 ± 0.814 | 4.648 ± 0.966 | 1.270 ± 0.808 | 0.266 ± 0.232 |
| MagicTelescope | 2.659 ± 1.637 | 0.473 ± 0.632 | 0.077 ± 0.110 | 2.012 ± 1.343 | 0.534 ± 0.565 | 0.093 ± 0.144 |
| MiniBooNE | 22.344 ± 7.001 | 2.388 ± 0.194 | 1.628 ± 0.208 | 16.454 ± 6.706 | 0.075 ± 0.086 | 0.012 ± 0.019 |
| bank-marketing | 13.512 ± 6.416 | 2.998 ± 1.582 | 0.925 ± 0.688 | 12.856 ± 6.609 | 2.324 ± 1.618 | 0.634 ± 0.433 |
| california | 8.281 ± 4.253 | 0.874 ± 0.524 | 0.351 ± 0.267 | 6.578 ± 4.264 | 0.342 ± 0.209 | 0.034 ± 0.024 |
| covertype | 23.977 ± 2.565 | 2.073 ± 0.657 | 0.976 ± 0.523 | 21.790 ± 2.253 | 0.992 ± 0.496 | 0.422 ± 0.319 |
| credit | 6.912 ± 4.083 | 2.369 ± 0.887 | 0.662 ± 0.606 | 5.739 ± 4.502 | 1.324 ± 0.674 | 0.350 ± 0.522 |
| default-of-credit-card-clients | 16.301 ± 4.462 | 4.512 ± 1.033 | 2.902 ± 0.620 | 7.618 ± 3.873 | 0.728 ± 0.331 | 0.531 ± 0.557 |
| electricity | 8.835 ± 1.824 | 1.060 ± 0.684 | 0.279 ± 0.266 | 7.952 ± 1.995 | 0.731 ± 0.383 | 0.285 ± 0.200 |
| eye_movements | 22.604 ± 1.486 | 12.687 ± 1.645 | 7.826 ± 1.822 | 2.884 ± 1.646 | 0.825 ± 0.711 | 0.607 ± 0.259 |
| heloc | 6.282 ± 2.351 | 2.517 ± 1.156 | 1.507 ± 0.498 | 1.625 ± 1.480 | 0.869 ± 0.957 | 0.727 ± 0.785 |
| house_16H | 13.600 ± 5.135 | 3.302 ± 0.376 | 1.950 ± 0.346 | 8.055 ± 4.429 | 0.330 ± 0.441 | 0.158 ± 0.098 |
| jannis | 19.390 ± 1.013 | 11.358 ± 0.377 | 7.140 ± 0.538 | 1.999 ± 1.237 | 0.305 ± 0.409 | 0.214 ± 0.235 |
| pol | 20.125 ± 2.902 | 5.059 ± 1.482 | 2.544 ± 1.005 | 15.887 ± 3.061 | 2.100 ± 1.358 | 0.751 ± 0.892 |

Table 9: Accuracy barrier for oblivious trees with AM.

| Dataset | Train | | | Test | | |
|---|---|---|---|---|---|---|
| | Naive | Perm | Perm&Order&Flip | Naive | Perm | Perm&Order&Flip |
| Bioresponse | 16.642 ± 4.362 | 19.033 ± 8.533 | 6.358 ± 1.915 | 7.165 ± 2.547 | 6.904 ± 5.380 | 1.038 ± 0.591 |
| Diabetes130US | 3.170 ± 3.304 | 5.473 ± 3.260 | 0.703 ± 0.517 | 2.831 ± 3.476 | 5.290 ± 3.486 | 0.390 ± 0.291 |
| Higgs | 28.640 ± 0.914 | 33.234 ± 3.164 | 15.678 ± 0.713 | 4.648 ± 0.966 | 8.113 ± 2.614 | 0.415 ± 0.454 |
| MagicTelescope | 2.659 ± 1.637 | 3.902 ± 1.931 | 0.224 ± 0.256 | 2.012 ± 1.343 | 3.687 ± 1.876 | 0.334 ± 0.434 |
| MiniBooNE | 22.344 ± 7.001 | 41.022 ± 3.398 | 2.184 ± 0.425 | 16.454 ± 6.706 | 34.452 ± 3.161 | 0.033 ± 0.056 |
| bank-marketing | 13.512 ± 6.416 | 12.248 ± 6.748 | 1.330 ± 0.806 | 12.856 ± 6.609 | 11.356 ± 7.168 | 0.695 ± 0.464 |
| california | 8.281 ± 4.253 | 9.539 ± 4.798 | 0.371 ± 0.365 | 6.578 ± 4.264 | 8.354 ± 4.648 | 0.112 ± 0.181 |
| covertype | 23.977 ± 2.565 | 27.590 ± 2.172 | 1.051 ± 0.407 | 21.790 ± 2.253 | 25.289 ± 1.787 | 0.403 ± 0.236 |
| credit | 6.912 ± 4.083 | 9.839 ± 6.698 | 1.169 ± 0.839 | 5.739 ± 4.502 | 8.291 ± 7.268 | 0.549 ± 0.751 |
| default-of-credit-card-clients | 16.301 ± 4.462 | 21.746 ± 7.075 | 3.646 ± 0.520 | 7.618 ± 3.873 | 12.183 ± 5.954 | 0.285 ± 0.372 |
| electricity | 8.835 ± 1.824 | 18.177 ± 5.979 | 0.472 ± 0.507 | 7.952 ± 1.995 | 17.396 ± 5.809 | 0.405 ± 0.356 |
| eye_movements | 22.604 ± 1.486 | 23.221 ± 3.024 | 8.588 ± 2.248 | 2.884 ± 1.646 | 2.761 ± 1.628 | 0.398 ± 0.435 |
| heloc | 6.282 ± 2.351 | 9.074 ± 3.894 | 2.541 ± 0.471 | 1.625 ± 1.480 | 3.891 ± 2.655 | 0.485 ± 0.397 |
| house_16H | 13.600 ± 5.135 | 17.963 ± 5.099 | 2.841 ± 0.543 | 8.055 ± 4.429 | 12.192 ± 4.635 | 0.292 ± 0.157 |
| jannis | 19.390 ± 1.013 | 22.482 ± 3.113 | 9.570 ± 0.316 | 1.999 ± 1.237 | 4.292 ± 2.509 | 0.069 ± 0.154 |
| pol | 20.125 ± 2.902 | 19.558 ± 5.785 | 3.056 ± 0.510 | 15.887 ± 3.061 | 14.858 ± 5.523 | 0.961 ± 0.722 |

Table 10: Accuracy barrier for decision lists with WM.

| Dataset | Train | | | | Test | | | |
|---|---|---|---|---|---|---|---|---|
| | Naive | Perm | Naive (Modified) | Perm (Modified) | Naive | Perm | Naive (Modified) | Perm (Modified) |
| Bioresponse | 21.323 ± 6.563 | 4.259 ± 0.698 | 14.578 ± 3.930 | 4.641 ± 0.918 | 9.325 ± 3.988 | 0.346 ± 0.277 | 7.346 ± 4.261 | 1.309 ± 0.827 |
| Diabetes130US | 5.182 ± 3.745 | 1.483 ± 1.006 | 2.754 ± 1.098 | 1.088 ± 0.608 | 4.910 ± 4.244 | 1.293 ± 1.332 | 1.476 ± 1.308 | 0.849 ± 0.885 |
| Higgs | 27.778 ± 1.036 | 16.110 ± 0.518 | 28.915 ± 1.314 | 14.071 ± 0.395 | 4.777 ± 0.803 | 0.106 ± 0.203 | 5.136 ± 0.946 | 0.039 ± 0.083 |
| MagicTelescope | 4.855 ± 3.388 | 0.355 ± 0.682 | 5.138 ± 2.655 | 0.182 ± 0.141 | 4.137 ± 3.763 | 0.280 ± 0.519 | 4.534 ± 2.588 | 0.157 ± 0.162 |
| MiniBooNE | 23.059 ± 1.479 | 1.911 ± 0.138 | 14.916 ± 3.616 | 1.580 ± 0.178 | 17.248 ± 1.683 | 0.025 ± 0.036 | 9.340 ± 3.585 | 0.035 ± 0.042 |
| bank-marketing | 11.952 ± 3.794 | 0.979 ± 0.478 | 11.589 ± 2.167 | 0.373 ± 0.448 | 11.387 ± 4.113 | 0.536 ± 0.472 | 10.540 ± 2.067 | 0.349 ± 0.348 |
| california | 6.522 ± 3.195 | 0.621 ± 0.363 | 8.435 ± 3.273 | 0.538 ± 0.214 | 5.167 ± 2.962 | 0.236 ± 0.146 | 6.844 ± 3.087 | 0.151 ± 0.147 |
| covertype | 13.408 ± 3.839 | 1.341 ± 0.433 | 11.114 ± 2.689 | 1.257 ± 0.904 | 11.162 ± 3.620 | 0.472 ± 0.340 | 8.826 ± 2.729 | 0.477 ± 0.889 |
| credit | 11.238 ± 8.115 | 1.968 ± 0.990 | 14.626 ± 5.448 | 1.390 ± 0.423 | 10.880 ± 9.040 | 1.421 ± 1.046 | 13.667 ± 5.951 | 0.940 ± 0.612 |
| default-of-credit-card-clients | 12.513 ± 5.116 | 3.107 ± 1.123 | 11.378 ± 2.123 | 3.793 ± 0.881 | 5.161 ± 4.304 | 0.328 ± 0.512 | 3.197 ± 1.916 | 0.666 ± 0.651 |
| electricity | 6.524 ± 1.863 | 0.725 ± 0.451 | 9.101 ± 2.685 | 0.944 ± 0.557 | 5.834 ± 1.838 | 0.420 ± 0.354 | 8.487 ± 2.460 | 0.543 ± 0.511 |
| eye_movements | 19.125 ± 1.791 | 9.433 ± 1.385 | 19.738 ± 1.490 | 8.755 ± 1.391 | 1.990 ± 1.623 | 0.329 ± 0.102 | 1.916 ± 1.492 | 0.277 ± 0.302 |
| heloc | 4.513 ± 1.826 | 1.564 ± 0.617 | 5.116 ± 0.793 | 1.574 ± 0.154 | 0.725 ± 0.598 | 0.155 ± 0.190 | 1.263 ± 0.711 | 0.359 ± 0.346 |
| house_16H | 9.195 ± 2.408 | 2.520 ± 0.446 | 8.693 ± 1.302 | 2.222 ± 0.730 | 4.629 ± 2.314 | 0.063 ± 0.129 | 4.192 ± 1.517 | 0.185 ± 0.296 |
| jannis | 20.766 ± 2.097 | 9.484 ± 0.371 | 20.520 ± 1.017 | 7.400 ± 0.324 | 3.947 ± 2.605 | 0.006 ± 0.013 | 4.451 ± 1.300 | 0.004 ± 0.009 |
| pol | 23.401 ± 5.448 | 3.137 ± 1.038 | 20.137 ± 4.200 | 3.435 ± 0.675 | 18.933 ± 5.249 | 0.952 ± 0.925 | 16.522 ± 3.502 | 1.143 ± 0.565 |

Table 11: Accuracy barrier for decision lists with AM.

| Dataset | Train | | | | Test | | | |
|---|---|---|---|---|---|---|---|---|
| | Naive | Perm | Naive (Modified) | Perm (Modified) | Naive | Perm | Naive (Modified) | Perm (Modified) |
| Bioresponse | 21.323 ± 6.563 | 13.349 ± 5.943 | 14.578 ± 3.930 | 10.363 ± 7.256 | 9.325 ± 3.988 | 4.817 ± 2.825 | 7.346 ± 4.261 | 3.871 ± 4.608 |
| Diabetes130US | 5.182 ± 3.745 | 5.590 ± 3.328 | 2.754 ± 1.098 | 1.371 ± 0.507 | 4.910 ± 4.244 | 4.926 ± 3.796 | 1.476 ± 1.308 | 0.694 ± 0.649 |
| Higgs | 27.778 ± 1.036 | 28.910 ± 2.132 | 28.915 ± 1.314 | 20.131 ± 1.693 | 4.777 ± 0.803 | 6.722 ± 1.231 | 5.136 ± 0.946 | 1.755 ± 1.403 |
| MagicTelescope | 4.855 ± 3.388 | 3.349 ± 3.273 | 5.138 ± 2.655 | 1.451 ± 0.705 | 4.137 ± 3.763 | 3.001 ± 3.478 | 4.534 ± 2.588 | 1.090 ± 0.437 |
| MiniBooNE | 23.059 ± 1.479 | 18.149 ± 7.500 | 14.916 ± 3.616 | 3.870 ± 1.168 | 17.248 ± 1.683 | 13.868 ± 7.222 | 9.340 ± 3.585 | 0.797 ± 0.860 |
| bank-marketing | 11.952 ± 3.794 | 9.782 ± 6.722 | 11.589 ± 2.167 | 2.815 ± 0.957 | 11.387 ± 4.113 | 9.151 ± 7.204 | 10.540 ± 2.067 | 2.521 ± 1.055 |
| california | 6.522 ± 3.195 | 5.812 ± 2.365 | 8.435 ± 3.273 | 2.254 ± 0.813 | 5.167 ± 2.962 | 4.899 ± 2.018 | 6.844 ± 3.087 | 1.186 ± 0.643 |
| covertype | 13.408 ± 3.839 | 14.727 ± 7.029 | 11.114 ± 2.689 | 4.036 ± 1.450 | 11.162 ± 3.620 | 13.352 ± 7.056 | 8.826 ± 2.729 | 2.656 ± 1.302 |
| credit | 11.238 ± 8.115 | 18.620 ± 9.806 | 14.626 ± 5.448 | 8.979 ± 6.919 | 10.880 ± 9.040 | 18.606 ± 10.015 | 13.667 ± 5.951 | 8.113 ± 6.633 |
| default-of-credit-card-clients | 12.513 ± 5.116 | 12.880 ± 5.070 | 11.378 ± 2.123 | 6.055 ± 1.178 | 5.161 ± 4.304 | 6.465 ± 5.062 | 3.197 ± 1.916 | 0.533 ± 0.239 |
| electricity | 6.524 ± 1.863 | 4.988 ± 2.732 | 9.101 ± 2.685 | 3.041 ± 0.676 | 5.834 ± 1.838 | 4.361 ± 2.532 | 8.487 ± 2.460 | 2.637 ± 0.730 |
| eye_movements | 19.125 ± 1.791 | 18.694 ± 1.774 | 19.738 ± 1.490 | 13.408 ± 1.196 | 1.990 ± 1.623 | 3.046 ± 1.625 | 1.916 ± 1.492 | 1.807 ± 1.312 |
| heloc | 4.513 ± 1.826 | 5.504 ± 1.650 | 5.116 ± 0.793 | 3.287 ± 0.758 | 0.725 ± 0.598 | 1.711 ± 1.278 | 1.263 ± 0.711 | 0.528 ± 0.147 |
| house_16H | 9.195 ± 2.408 | 8.591 ± 3.370 | 8.693 ± 1.302 | 3.937 ± 0.816 | 4.629 ± 2.314 | 4.547 ± 2.726 | 4.192 ± 1.517 | 0.751 ± 0.508 |
| jannis | 20.766 ± 2.097 | 20.768 ± 2.200 | 20.520 ± 1.017 | 12.008 ± 0.892 | 3.947 ± 2.605 | 6.472 ± 2.342 | 4.451 ± 1.300 | 0.106 ± 0.162 |
| pol | 23.401 ± 5.448 | 17.384 ± 6.441 | 20.137 ± 4.200 | 10.339 ± 2.743 | 18.933 ± 5.249 | 13.285 ± 5.863 | 16.522 ± 3.502 | 6.492 ± 2.536 |

Table 12: Training accuracy barrier for permuted models with WM. The numbers in parentheses represent the original accuracy.

| Dataset | Non-Oblivious Tree | Oblivious Tree | Decision List | Decision List (Modified) |
|---|---|---|---|---|
| Bioresponse | 5.876 ± 1.477 (93.005) | 4.800 ± 0.895 (91.753) | 4.259 ± 0.698 (91.771) | 4.641 ± 0.918 (90.489) |
| Diabetes130US | 1.388 ± 1.159 (60.686) | 1.120 ± 1.123 (60.567) | 1.483 ± 1.006 (60.425) | 1.088 ± 0.608 (61.178) |
| Higgs | 18.470 ± 0.769 (97.232) | 19.754 ± 1.023 (97.616) | 16.110 ± 0.518 (95.838) | 14.071 ± 0.395 (95.831) |
| MagicTelescope | 0.576 ± 0.556 (84.963) | 0.473 ± 0.632 (84.460) | 0.355 ± 0.682 (84.999) | 0.182 ± 0.141 (85.411) |
| MiniBooNE | 2.272 ± 0.215 (99.980) | 2.388 ± 0.194 (99.980) | 1.911 ± 0.138 (99.977) | 1.580 ± 0.178 (99.976) |
| bank-marketing | 2.711 ± 1.183 (79.490) | 2.998 ± 1.582 (79.351) | 0.979 ± 0.478 (79.166) | 0.373 ± 0.448 (79.709) |
| california | 0.873 ± 0.551 (87.897) | 0.874 ± 0.524 (87.909) | 0.621 ± 0.363 (88.012) | 0.538 ± 0.214 (88.054) |
| covertype | 1.839 ± 0.336 (79.445) | 2.073 ± 0.657 (79.754) | 1.341 ± 0.433 (79.618) | 1.257 ± 0.904 (79.550) |
| credit | 3.172 ± 2.636 (78.679) | 2.369 ± 0.887 (78.231) | 1.968 ± 0.990 (78.166) | 1.390 ± 0.423 (78.905) |
| default-of-credit-card-clients | 5.419 ± 1.318 (78.017) | 4.512 ± 1.033 (78.657) | 3.107 ± 1.123 (77.315) | 3.793 ± 0.881 (78.308) |
| electricity | 1.035 ± 0.543 (80.375) | 1.060 ± 0.684 (80.861) | 0.725 ± 0.451 (80.396) | 0.944 ± 0.557 (80.651) |
| eye_movements | 11.605 ± 1.927 (81.693) | 12.687 ± 1.645 (83.730) | 9.433 ± 1.385 (81.075) | 8.755 ± 1.391 (81.451) |
| heloc | 1.652 ± 0.475 (77.430) | 2.517 ± 1.156 (78.370) | 1.564 ± 0.617 (77.968) | 1.574 ± 0.154 (78.550) |
| house_16H | 3.362 ± 0.482 (93.093) | 3.302 ± 0.376 (93.351) | 2.520 ± 0.446 (92.783) | 2.222 ± 0.730 (93.058) |
| jannis | 10.442 ± 1.404 (100.000) | 11.358 ± 0.377 (100.000) | 9.484 ± 0.371 (100.000) | 7.400 ± 0.324 (100.000) |
| pol | 4.612 ± 0.912 (98.348) | 5.059 ± 1.482 (98.340) | 3.137 ± 1.038 (97.883) | 3.435 ± 0.675 (97.881) |

Table 13: Training accuracy barrier for permuted models with AM. The numbers in parentheses represent the original accuracy.

| Dataset | Non-Oblivious | Oblivious | Decision List | Decision List (Modified) |
|---|---|---|---|---|
| Bioresponse | 14.066 ± 7.045 (93.005) | 19.033 ± 8.533 (91.753) | 13.349 ± 5.943 (91.771) | 10.363 ± 7.256 (90.489) |
| Diabetes130US | 3.086 ± 2.566 (60.686) | 5.473 ± 3.260 (60.567) | 5.590 ± 3.328 (60.425) | 1.371 ± 0.507 (61.178) |
| Higgs | 30.704 ± 2.899 (97.232) | 33.234 ± 3.164 (97.616) | 28.910 ± 2.132 (95.838) | 20.131 ± 1.693 (95.831) |
| MagicTelescope | 3.309 ± 1.486 (84.963) | 3.902 ± 1.931 (84.460) | 3.349 ± 3.273 (84.999) | 1.451 ± 0.705 (85.411) |
| MiniBooNE | 34.934 ± 8.157 (99.980) | 41.022 ± 3.398 (99.980) | 18.149 ± 7.500 (99.977) | 3.870 ± 1.168 (99.976) |
| bank-marketing | 13.598 ± 7.638 (79.490) | 12.248 ± 6.748 (79.351) | 9.782 ± 6.722 (79.166) | 2.815 ± 0.957 (79.709) |
| california | 5.800 ± 2.036 (87.897) | 9.539 ± 4.798 (87.909) | 5.812 ± 2.365 (88.012) | 2.254 ± 0.813 (88.054) |
| covertype | 19.708 ± 6.392 (79.445) | 27.590 ± 2.172 (79.754) | 14.727 ± 7.029 (79.618) | 4.036 ± 1.450 (79.550) |
| credit | 10.556 ± 8.753 (78.679) | 9.839 ± 6.698 (78.231) | 18.620 ± 9.806 (78.166) | 8.979 ± 6.919 (78.905) |
| default-of-credit-card-clients | 14.166 ± 2.297 (78.017) | 21.746 ± 7.075 (78.657) | 12.880 ± 5.070 (77.315) | 6.055 ± 1.178 (78.308) |
| electricity | 12.955 ± 4.558 (80.375) | 18.177 ± 5.979 (80.861) | 4.988 ± 2.732 (80.396) | 3.041 ± 0.676 (80.651) |
| eye_movements | 18.757 ± 1.273 (81.693) | 23.221 ± 3.024 (83.730) | 18.694 ± 1.774 (81.075) | 13.408 ± 1.196 (81.451) |
| heloc | 6.564 ± 2.404 (77.430) | 9.074 ± 3.894 (78.370) | 5.504 ± 1.650 (77.968) | 3.287 ± 0.758 (78.550) |
| house_16H | 10.184 ± 2.667 (93.093) | 17.963 ± 5.099 (93.351) | 8.591 ± 3.370 (92.783) | 3.937 ± 0.816 (93.058) |
| jannis | 19.004 ± 1.246 (100.000) | 22.482 ± 3.113 (100.000) | 20.768 ± 2.200 (100.000) | 12.008 ± 0.892 (100.000) |
| pol | 16.267 ± 3.914 (98.348) | 19.558 ± 5.785 (98.340) | 17.384 ± 6.441 (97.883) | 10.339 ± 2.743 (97.881) |

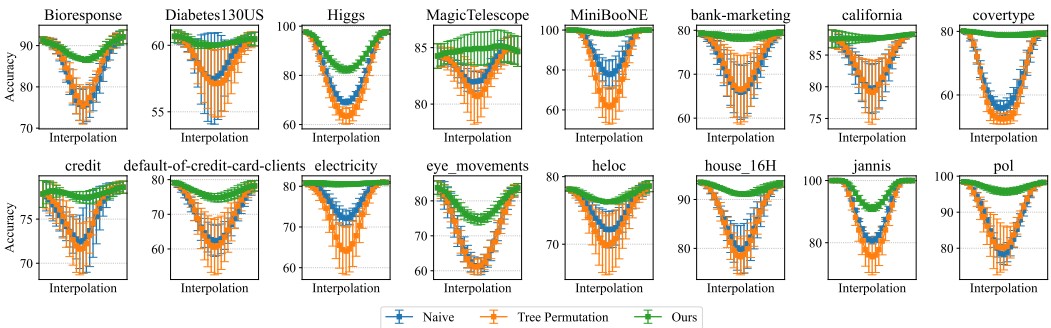

Figure 9: Interpolation curves of train accuracy for oblivious trees with AM.

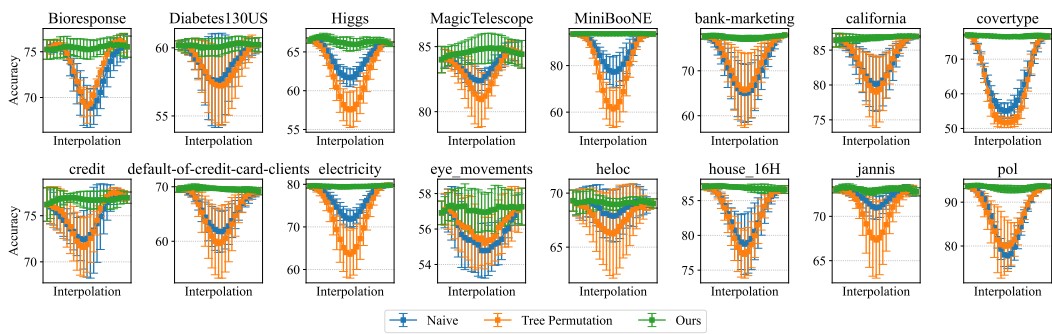

Figure 10: Interpolation curves of test accuracy for oblivious trees with AM.

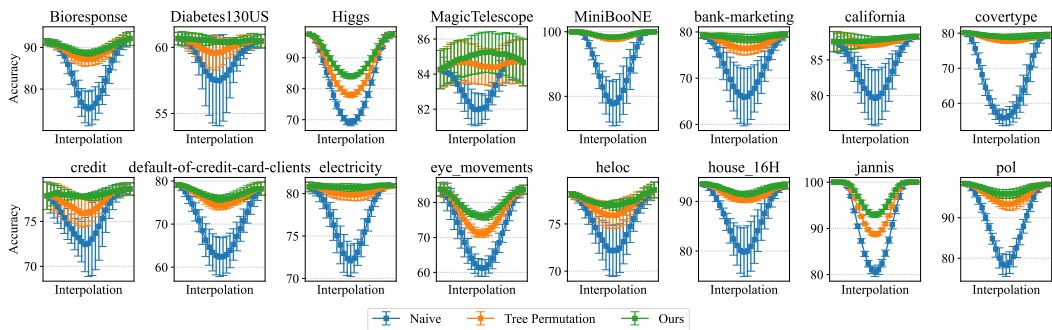

Figure 11: Interpolation curves of train accuracy for oblivious trees with WM.

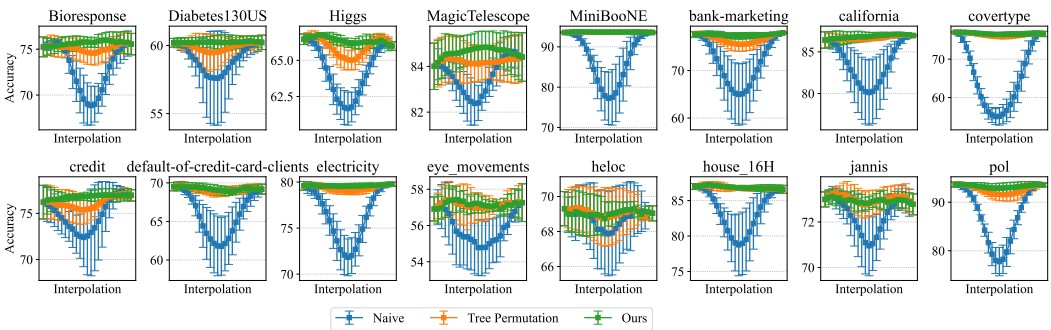

Figure 12: Interpolation curves of test accuracy for oblivious trees with WM.

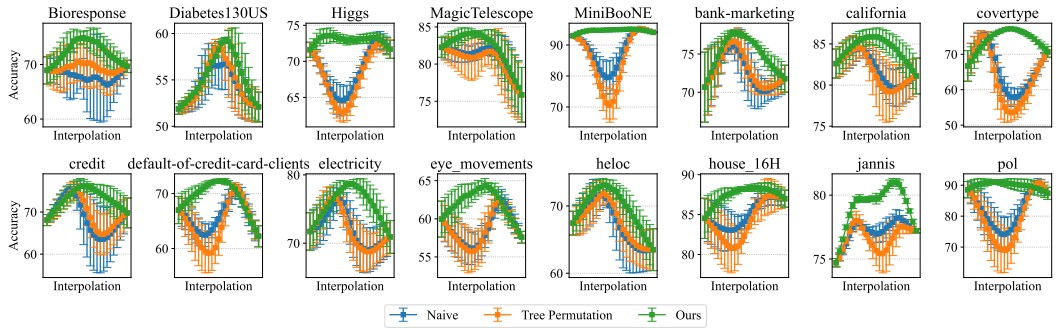

Figure 13: Interpolation curves of train accuracy for oblivious trees with AM by use of split dataset.

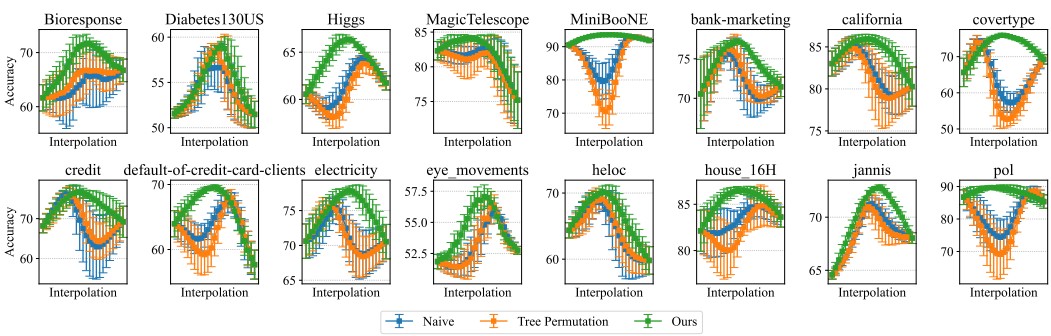

Figure 14: Interpolation curves of test accuracy for oblivious trees with AM by use of split dataset.

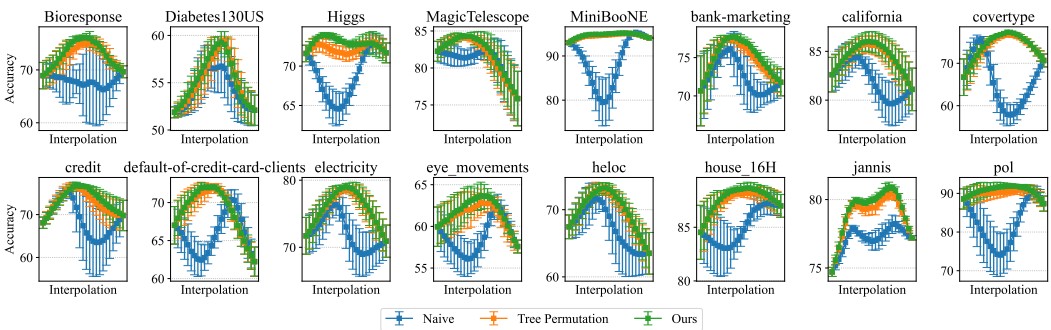

Figure 15: Interpolation curves of train accuracy for oblivious trees with WM by use of split dataset.

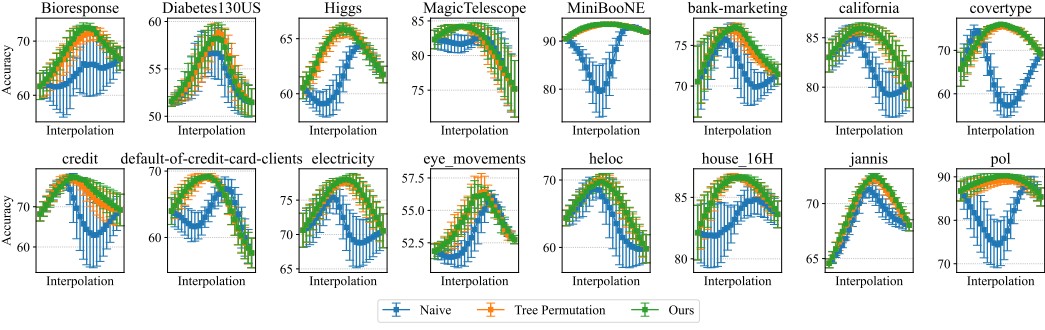

Figure 16: Interpolation curves of test accuracy for oblivious trees with WM by use of split dataset.

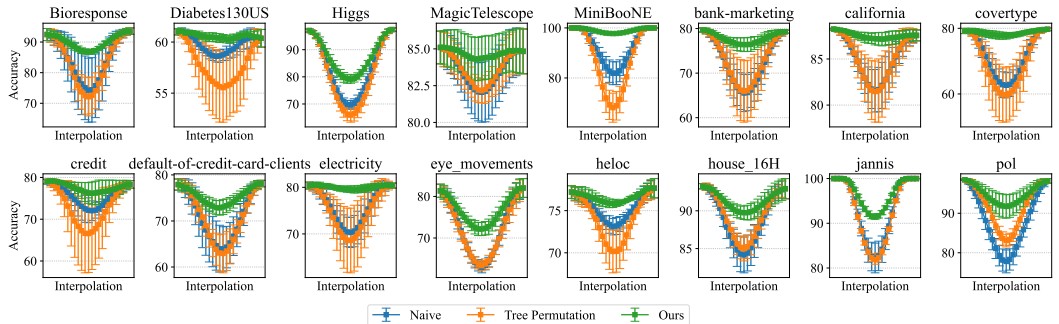

Figure 17: Interpolation curves of train accuracy for non-oblivious trees with AM.

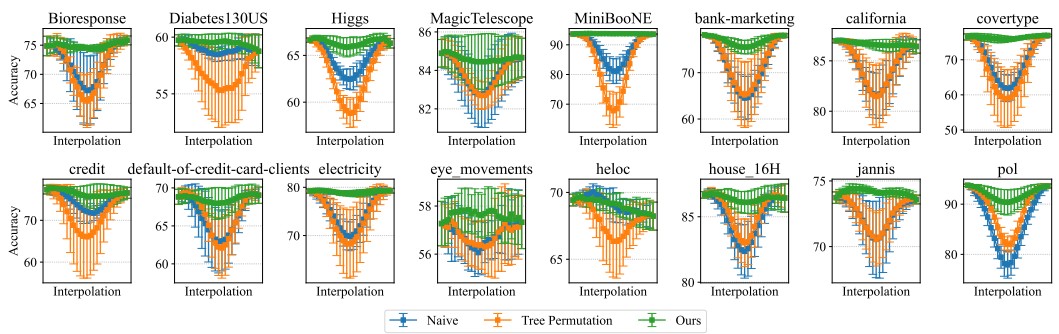

Figure 18: Interpolation curves of test accuracy for non-oblivious trees with AM.

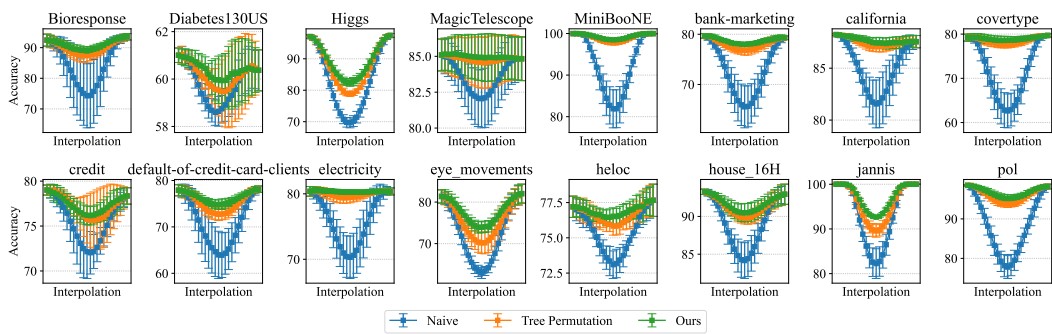

Figure 19: Interpolation curves of train accuracy for non-oblivious trees with WM.

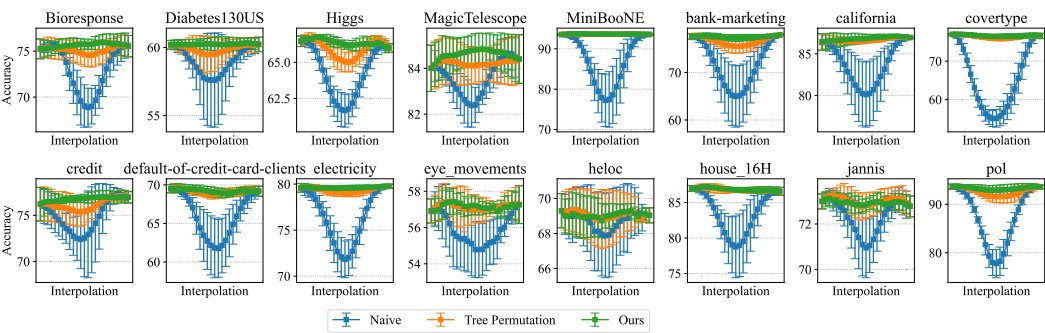

Figure 20: Interpolation curves of test accuracy for non-oblivious trees with WM.

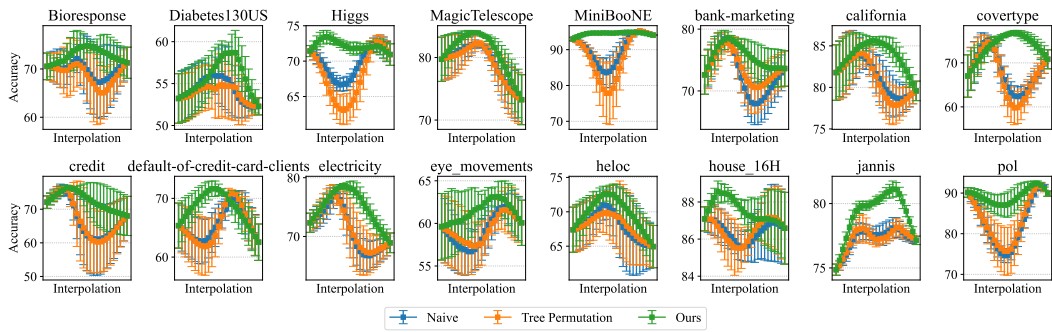

Figure 21: Interpolation curves of train accuracy for non-oblivious trees with AM by use of split dataset.

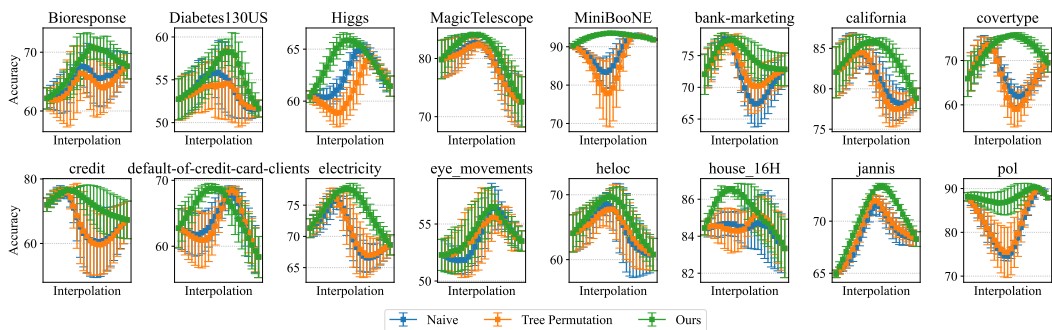

Figure 22: Interpolation curves of test accuracy for non-oblivious trees with AM by use of split dataset.

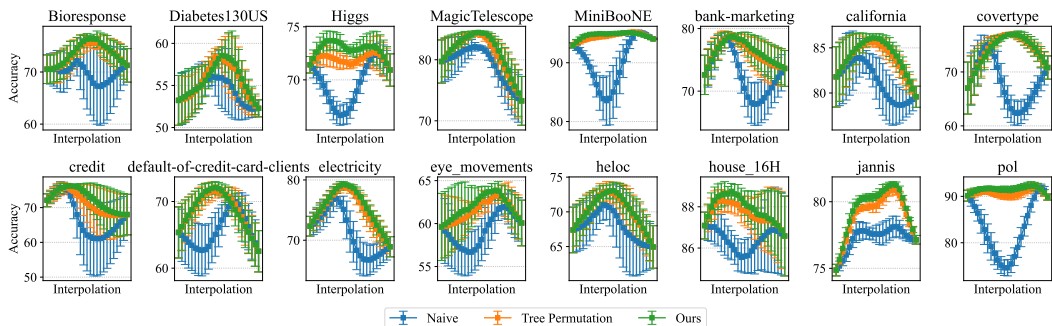

Figure 23: Interpolation curves of train accuracy for non-oblivious trees with WM by use of split dataset.

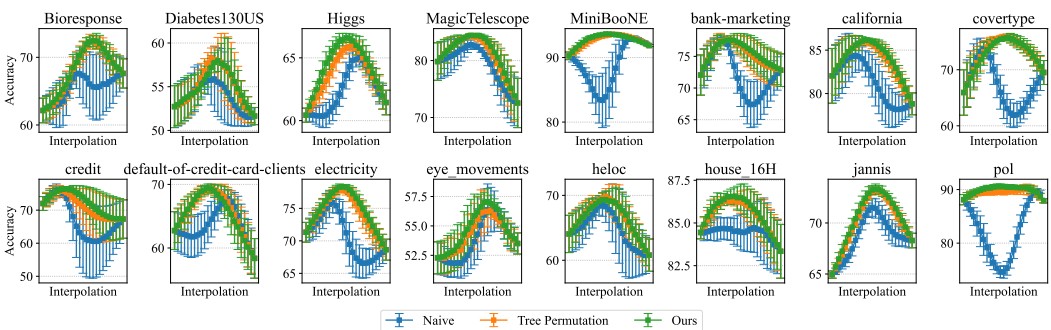

Figure 24: Interpolation curves of test accuracy for non-oblivious trees with WM by use of split dataset.

