# OpenReview forum: "Linear Mode Connectivity in Differentiable Tree Ensembles"
_NeurIPS.cc/2024/Conference — Submitted to NeurIPS 2024_

### Official Review · Reviewer_Varw · 2024-07-06

**Soundness:** 4
**Presentation:** 3
**Contribution:** 3
**Rating:** 7
**Confidence:** 4

**Summary:**

This work extrapolates the concept of Linear Mode Connectivity (LMC) modulo model invariances to differentiable tree ensembles (DTE). The authors revealed that, in contrast to neural networks (NNs), permutation invariance is insufficient to provide LMC in DTE and propose two additional tree-specific invariances that enable LMC after taking them into account: subtree flip invariance and splitting order invariance. In addition, they provide a modified DTE architecture that does not posses these additional invariances, however still enjoys LMC with only permutation invariance akin to neural network models. This work proposes two algorithms for building LMC given two independently trained DTEs, based on similar methods from NN LMC literature. The claims are supported by a detailed empirical evaluation.

**Strengths:**

Honestly, I enjoyed reading this paper. Although I am not specialized in tree ensembles, I have certain expertise in LMC, and was pleased to find that it is also relevant for DTE models. I think that this contribution is novel and significant.

The paper is very well-structured. It was very easy to follow despite having no significant experience in decision trees, the authors did a good job preparing the reader in Sec. 2.

Section 3 presents the main contributions of this work, which is done very well using both detailed and intuitive text description and auxiliary images illustrating the main concepts.

Empirical evaluation is excellent, involving multiple datasets, hyperparameter options, and random seeds. The authors tackled many important questions concerning the study of LMC in DTEs and even compared with NN LMC, which I specifically liked.

**Weaknesses:**

It is hard for me to formulate substantial flaws in this work but a couple of remarks that I put in the next section.

The main weakness of this work is lack of theoretical support and practical implications. However, I acknowledge that these are the same limitations that are attributed to LMC in neural networks, which is a significantly more broad and well-studied field than LMC in tree ensembles. I hope that future work will address these disadvantages in some way.

Also, I believe that the text could be slightly polished to eliminate typos and small inaccuracies. For instance, the value $D$ in line 127 is not defined at its first occurrence.

**Questions:**

Below I list some questions/comments related to the work.
1. It is indeed surprising that mode connectivity can worsen compared to a naive interpolation after accounting for the permutation invariance (e.g., Fig. 1). To my view, the barriers of naive interpolation (realizing an identity permutation) must upper bound the barriers after permutation search. I would ask the authors to give a small comment on this.
2. Is the weighting strategy described in Sec. 3.2 remains the same for oblivious DTEs that share the same parameters for all nodes at equal depth, so every parameter affects all leaves?
3. Interestingly, according to Figure 5, non-oblivious DTE shows better LMC than oblivious DTE when accounting for the same invariances. I suppose that this benign over-parameterization effect is similar to increasing $M$ in DTE or width in NN. What do the authors think?
4. Figure 7: I would suggest adding the result of common (combined data) training for comparison alike [1].
5. Does LMC in DTE suffer from the same variance collapse issues reported by [2]? If do, how could it be repaired and could it additionally improve LMC in DTE?
6. It is a little astonishing how much worse Activation Matching performs in DTE compared to Weight Matching. In NN LMC (based on literature and my personal experience), this difference is not so prominent. Could the authors comment on this a little more?
7. At the same depth, (modified) decision lists contain less total parameters than oblivious trees and especially non-oblivious trees. However, all DTE architectures perform very similarly according to Tab. 3 and 4. Could the authors give any explanation of this effect?
8. Would adding subtree flip invariance in the terminal splitting node of regular decision lists improve their LMC reported in Tab. 3 and 4?

[1] Samuel Ainsworth, Jonathan Hayase, and Siddhartha Srinivasa. Git Re-Basin: Merging Models modulo Permutation Symmetries. In The Eleventh International Conference on Learning Representations, 2023.

[2] Keller Jordan, Hanie Sedghi, Olga Saukh, Rahim Entezari, and Behnam Neyshabur. REPAIR: REnormalizing permuted activations for interpolation repair. In The Eleventh International Conference on Learning Representations, 2023.

**Limitations:**

The authors discuss the limitations of their methods in Section 3.2.

---

> ### Author Rebuttal · Authors · 2024-08-04
>
> Thank you for your review.
>
> > The main weakness of this work is lack of theoretical support and practical implications. However, I acknowledge that these are the same limitations that are attributed to LMC in neural networks, which is a significantly more broad and well-studied field than LMC in tree ensembles. I hope that future work will address these disadvantages in some way.
>
> Thank you for your understanding.
>
> > Also, I believe that the text could be slightly polished to eliminate typos and small inaccuracies. For instance, the value in line 127 is not defined at its first occurrence.
>
> Sorry, $D$ refers to the depth of the tree. This will be corrected in the camera-ready version.
>
> > It is indeed surprising that mode connectivity can worsen compared to a naive interpolation after accounting for the permutation invariance (e.g., Fig. 1). To my view, the barriers of naive interpolation (realizing an identity permutation) must upper bound the barriers after permutation search. I would ask the authors to give a small comment on this.
>
> Since we are only looking at the similarity of activations and weights, it does not necessarily mean there will be an improvement in the loss landscape. For example, in Figure 1, even though the distance between parameters is smaller after considering permutations (top right model to the Target), the barrier is larger compared to the naive interpolation from the Origin to the Target (bottom left to the Target). This might be the situation occurring here.
>
> > Is the weighting strategy described in Sec. 3.2 remains the same for oblivious DTEs that share the same parameters for all nodes at equal depth, so every parameter affects all leaves?
>
> Yes, your understanding is correct. In the case of an oblivious tree, the contribution of all splitting rules is equal.
>
> > Interestingly, according to Figure 5, non-oblivious DTE shows better LMC than oblivious DTE when accounting for the same invariances. I suppose that this benign over-parameterization effect is similar to increasing in DTE or width in NN. What do the authors think?
>
> I believe the reason is that non-oblivious trees exhibit a greater number of invariance patterns, which increases the likelihood of achieving LMC through parameter transformation. When considering a tree of depth $D$ , the number of subtree flip invariance patterns in an oblivious tree is $2^D$, whereas for a non-oblivious tree, it is $2^{2^{D}-1}$. As you mentioned, this could be seen as a benefit of overparameterization.
>
> > Figure 7: I would suggest adding the result of common (combined data) training for comparison alike [1].
>
> Thank you for your comment. Please check our uploaded PDF in the rebuttal. Through model merging, it demonstrates similar performance to full data training even with split data training.
>
> > Does LMC in DTE suffer from the same variance collapse issues reported by [2]? If do, how could it be repaired and could it additionally improve LMC in DTE?
>
> Although this study does not investigate deeply into the topic, Equation (5) suggests a similarity between tree ensembles and neural networks. Therefore, techniques such as REPAIR could potentially enhance matching performance. Our research can be integrated with existing studies, including those on matching algorithms.
>
> > It is a little astonishing how much worse Activation Matching performs in DTE compared to Weight Matching. In NN LMC (based on literature and my personal experience), this difference is not so prominent. Could the authors comment on this a little more?
>
> In activation matching, we use the output of each individual tree, but the number of parameters required to obtain the output of each tree is greater compared to typical MLP activation matching. In MLPs, we only need to consider the parameters corresponding to the width of each layer for calculating an activation, but the tree structure adds complexity. This likely makes the matching problem more challenging in our case.
>
> > At the same depth, (modified) decision lists contain less total parameters than oblivious trees and especially non-oblivious trees. However, all DTE architectures perform very similarly according to Tab. 3 and 4. Could the authors give any explanation of this effect?
>
> Considering both representational capacity and training dynamics, such an outcome is possible. Additionally, since Table 3 and Table 4 consider trees of depth 2, the change in the number of parameters might not be significant, which could be another reason. Given the consistent results of other barrier-related experiments, it does not appear to be due to an implementation error (the reproducible code is also provided).
>
> > Would adding subtree flip invariance in the terminal splitting node of regular decision lists improve their LMC reported in Tab. 3 and 4?
>
> There is a possibility of improvement. However, the impact of this invariance is likely to be less significant compared to a perfect binary tree, so the performance gain may be smaller.

---

> > ### Comment · Reviewer_Varw · 2024-08-10
> > **Reviewer's response**
> >
> > Thanks a lot for the clarification and interesting discussion!
> > I especially appreciate the additional conducted experiments. It is indeed intriguing that in the case of DTE, model merging can reach the level of full data training.
> >
> > I personally like this work a lot and recommend acceptance, not changing my score.

---

> > > ### Author Response · Authors · 2024-08-12
> > >
> > > Thank you for your response and for supporting the acceptance. We are also pleased that we had fruitful discussions.

---

### Official Review · Reviewer_fRZd · 2024-07-12

**Soundness:** 2
**Presentation:** 3
**Contribution:** 2
**Rating:** 5
**Confidence:** 4

**Summary:**

This paper provides an analysis of types of neural networks called soft trees from the linear mode connectivity point of view. The authors enumerate 3 types of invariances inherent to soft trees and study linear mode connectivity between different solutions (by solution they understand a trained ensemble of soft tree models) after weights or activations matching that account for these invariances. They also study linear mode connectivity for a special case of soft trees - decision list-based tree - that has only one type of invariance.

**Strengths:**

- The paper is well written
- Authors claim that it is the first paper to study linear mode connectivity for soft trees

**Weaknesses:**

## Insufficient contribution
- In my opinion, the main contribution of this paper is a showcase that different architectures need to account for different invariances when LMC is analyzed, e.g. MLP and soft trees have different invariances. I think that this insight alone is not enough for a paper, because it sounds quite obvious even without analysis.

## Questionable results
- It is very important to make sure that interpolation results are not computed between the models which are almost identical (that can happen if there is not enough diversity in training recipes). Could you please provide results with distances (any kind of them, e.g. L2 or cosine similarity) between the solutions in Figure. 5 for "Naive", "Tree Permutation" and "Ours" parameter transformations?
- I would expect decision list trees to be much weaker than soft trees because they have less parameters. Could you please report its performance or show me where I can find it?
- Model merging is mentioned as one of the applications for linear mode connectivity (LMC), however, no results for model merging are provided.
  - line 32: "In addition, LMC also holds significant practical importance, enabling techniques such as model merging [6, 7] by weight-space parameter averaging."

## Questionable explanation
- I could not find a related work section.
- What is "Ours" in Table 2?
- I did not find in the main text any explanation (even after looking into algorithms in appendix, which I found very confusing) for the operation of weights matching (WM) and activation matching (AM) in case of such invariances as "Perm", "Order" and "Flip" (Notation is from Table 1). Since invariances are the main part of the whole analysis, could you please elaborate more?
- Another important part of parameter transforms includes Linear Assignment Problem (LAP), but I could not find any details for it neither.

**Questions:**

- What do you mean by crucial for the stable success of non-convex optimization? Since this motivates your analysis, could you justify the main text, please?
  - line 4: "considered crucial for validating the stable success of the non-convex optimization"
  - line 30: “From a theoretical perspective, LMC is crucial for supporting the stable and successful application of non-convex optimization.”
- Why does an MLP with depth 2 have lower accuracy than an MLP with depth 1 in Table 2?
- Could you please explain or provide sources for how soft trees help in the development of LLMs?
  - line 44: "contributes to broader research into essential technological components critical for the development of large-scale language models”
- Why is the accuracy of the interpolated model higher than the accuracy of the starting models in Figure 8 (e.g. for Bioresponse)?
- Why do you say that if models are LMC then they are functionally equivalent? Doesn't it just mean that they are connected with a path in weights space along which loss value is lower than some threshold?
  - line 29: “This demonstrates that the trained models reside in different, yet functionally equivalent, local minima. This situation is referred to as Linear Mode Connectivity (LMC) [5].”
- What is the sum of leaf parameters? Doesn't leaf have only one parameter? And what is its shape? According to eq. (3) it should be vector, I think.
  - line 119: "as sum of the leaf parameters $\pi_{m, \ell}$"
- Why do you study LMC between ensembles of trees and not single trees? That would remove the need for accounting for permutation invariance.
- Could you please explain why decision boundaries of oblivious trees are straight lines?
  - line 152: "which means that the decision boundaries are straight lines without any bends."

**Limitations:**

- There is no theoretical justification for why and in which scenarios linear mode connectivity exists for soft trees.
- The paper does not propose any practical application for the linear mode connectivity between soft trees. While it can be argued that this paper is an analysis paper, some practical applications can be useful in motivating this kind of analysis.
- I did not find the code of the project while in the survey it is written that code is provided in supplementary material.

---

> ### Author Rebuttal · Authors · 2024-08-04
>
> Thank you for your review.
>
> Due to the 6000 character limit, we will address minor points during the discussion phase. Below in the rebuttal, we have included responses to the aspects we consider important.
>
> > ​​In my opinion, the main contribution of this paper is a showcase that different architectures need to account for different invariances when LMC is analyzed, e.g. MLP and soft trees have different invariances. I think that this insight alone is not enough for a paper, because it sounds quite obvious even without analysis.
>
> We would like to emphasize that even if invariances exist, it is non-trivial whether they have an impact on LMC, and investigating it could provide a valuable contribution to the community. For example, ReLU networks exhibit permutation invariance and scaling invariance, but scaling invariance is not important for achieving LMC. While it was known that permutation invariance exists in neural networks, ascertaining whether its consideration could achieve LMC was a challenging question. Research investigating this question has had a significant impact on the community [1]. Moreover, architectures like transformers have different modules, such as attention mechanisms, and demonstrating the importance of considering these architecture-inherent invariances for matching has been highly valued and accepted at a recent conference [2].
>
> Our research contributes new insights regarding the existence of invariances and demonstrates that considering them can indeed achieve LMC. Additionally, we have shown that by adjusting the tree structure, we can optimize both the amount of invariance and computational efficiency, which is an essential consideration for practical applications and this idea can be applied to other model structures as well. We believe these findings are valuable and bring novel aspects to the community.
>
> > It is very important to make sure that interpolation results are not computed between the models which are almost identical (that can happen if there is not enough diversity in training recipes). Could you please provide results with distances (any kind of them, e.g. L2 or cosine similarity) between the solutions in Figure. 5 for "Naive", "Tree Permutation" and "Ours" parameter transformations?
>
> Thank you for your suggestion. We conducted an additional experiment using the MiniBooNE dataset, as referenced in Figure 1. The experimental settings are the same as those used for creating Figures 1 and 6.
>
> L2 Distances:
> - Naive: 98.01
> - Tree Permutation: 96.13
> - Ours: 78.04
>
> Cosine Similarity:
> - Naive: 0.0035
> - Tree Permutation: 0.0414
> - Ours: 0.3682
>
> These results indicate that the models are not nearly identical, as the distances do not approach zero even after matching. Therefore, we do not believe we are facing the issue you are concerned about. Additionally, it is known that achieving LMC does not strictly require the distances to be exactly zero [3].
>
> > What do you mean by crucial for the stable success of non-convex optimization? Since this motivates your analysis, could you justify the main text, please?
>
> Despite the challenges of non-convex optimization, our machine-learning community empirically observes that models consistently achieve similar performance even with different random initializations. This phenomenon is quite non-trivial, and understanding the underlying reasons is crucial. Achieving LMC implies that the solutions attained by training from different initial values are fundamentally equivalent. This suggests that the abundance of functional invariance is one of the reasons for the stable success of non-convex optimization. This perspective is also highlighted as a motivation in existing LMC research [1] and is mentioned in our introduction.
>
> > Why do you study LMC between ensembles of trees and not single trees? That would remove the need for accounting for permutation invariance.
>
> We can consider a single tree. However, as shown in previous studies [1] and Figure 5, the large number of trees (or the large width of neural networks) is known to be important for achieving LMC. Therefore, LMC is less likely to be achieved when considering only a single tree.
>
> > There is no theoretical justification for why and in which scenarios linear mode connectivity exists for soft trees.
>
> You are correct; the current community lacks a strong theoretical explanation for why LMC is achieved in neural networks, let alone in tree ensembles. Reviewer Varw has mentioned this perspective, and I hope you can check his/her comment: `The main weakness of this work is lack of theoretical support and practical implications. However, I acknowledge that these are the same limitations that are attributed to LMC in neural networks, which is a significantly more broad and well-studied field than LMC in tree ensembles. I hope that future work will address these disadvantages in some way. `.
>
> As shown in equation (5), soft tree ensembles and MLPs share fundamental similarities. Therefore, a deeper understanding of soft tree ensembles could also lead to a better understanding of neural networks.
>
> ----
>
> [1] Ainsworth et al., Git Re-Basin: Merging Models modulo Permutation Symmetries, ICLR2023
>
> [2] Imfeld et al., Transformer Fusion with Optimal Transport, ICLR2024
>
> [3] Ito et al., Analysis of Linear Mode Connectivity via Permutation-Based Weight Matching, arXiv 2402.04051

---

> > ### Author Response · Authors · 2024-08-07
> >
> > The responses to the minor points are as follows:
> >
> > > I would expect decision list trees to be much weaker than soft trees because they have less parameters.
> >
> > You can check it in Tables 3 and 4. Performance differences between perfect binary trees and decision lists are small.
> >
> > > Model merging is mentioned as one of the applications for linear mode connectivity (LMC), however, no results for model merging are provided
> >
> > Figure 7 shows the results. It can be observed that performance has improved through merging.
> >
> > > I could not find a related work section
> >
> > We have not explicitly separated a section for related work, but we do discuss related work in the introduction and conclusion. If it is necessary to have a distinct section, we can address this in the camera-ready version.
> >
> > > What is "Ours" in Table 2?
> >
> > “Ours” in Table 2 refers to the results when considering not only tree permutation invariance but also subtree flip invariance and splitting order invariance for matching. In Section 3, our method is mentioned, and the term “Ours” is also used in Figures 1, 6, and 7.
> >
> > > I did not find in the main text any explanation (even after looking into algorithms in appendix, which I found very confusing) for the operation of WM and AM...
> >
> > As mentioned, detailed explanations are provided in the appendix. We also explained in line 202 in the main text. If the absence of algorithmic details (Algorithms 1 and 2) in the main text lowers the evaluation, we can move them to the main part. However, we have structured the content this way to maximize the information within the page limit.
> >
> > > could not find any details for LAP
> >
> > Are you requesting a detailed explanation of LAP? If needed, we can add it to the appendix in the camera-ready version. We assumed that the LAP is well understood within the community, which is why previous representative studies, such as [1], did not explicitly explain it. Please note that we mention the algorithms to solve LAP (Jonker-Volgendant) in the main text.
> >
> > > Why does an MLP with depth 2 have lower accuracy than an MLP with depth 1 in Table 2?
> >
> > This is because Table 2 presents the generalization error. Generalization performance does not necessarily improve as the model becomes more complex.
> >
> > > Could you please explain or provide sources for how soft trees help in the development of LLMs?
> >
> > A soft tree can be interpreted as a hierarchical mixture of experts [4]. The mixture of experts is a technique used in large language models like Mistral [5]. While tree ensembles might not be directly used in LLM development, considering a scenario where each expert module in a hierarchical mixture of experts is a language model, subtree flip invariance and splitting order invariance become important when performing model merging.
> >
> > > Why is the accuracy of the interpolated model higher than the accuracy of the starting models?
> >
> > As shown in Figure 1, the line segments connecting the models after matching can often result in better performance in terms of generalization error. This phenomenon frequently occurs in experiments involving split data training [1] and has been observed in models like MLPs, not just tree ensembles.
> >
> > > Why do you say that if models are LMC then they are functionally equivalent? Doesn't it just mean that they are connected with a path in weights space along which loss value is lower than some threshold?
> >
> > Yes, since the barrier is not strictly zero in practice, your expression is more precise. If we consider a threshold to be zero, it would be equivalent to reaching the same local solution, achieving functional equivalence in such a case.
> >
> > > What is the sum of leaf parameters?
> >
> > Each leaf has a vector whose length equals the number of classes (this information is explicitly mentioned only in the function arguments on line 119, so we will clarify it in camera-ready). The term "parameters" might have caused some misunderstanding by suggesting it was a scalar. The model output is a weighted sum of the vector with a length equal to the number of classes, as shown in Equation (3).
> >
> > > why decision boundaries of oblivious trees are straight lines?
> >
> > In an oblivious tree, the splitting rules at the same depth share the same decision criteria, which include the slope and the intercept of the decision boundary. This means that regardless of the root-to-leaf path, the data passes through the same splitting rules, resulting in straight decision boundaries.
> >
> > > The paper does not propose any practical application for the linear mode connectivity between soft trees
> >
> > Model merging can be considered a practical application. Model merging has potential applications in continual learning [6] and federated learning [7].
> >
> > ----
> >
> > [4] Jordan and Jacobs, Hierarchical mixtures of experts and the EM algorithm, ICNN1993
> >
> > [5] Jiang et al., Mistral 7B, 2023
> >
> > [6] Mirzadeh et al., Linear Mode Connectivity in Multitask and Continual Learning, ICLR2021
> >
> > [7] Adilova et al., Layer-wise linear mode connectivity, ICLR2024

---

> ### Comment · Reviewer_fRZd · 2024-08-11
>
> Thank you for your rebuttal, it clarified some of my questions. I will keep my score and here is my justification for it:
>
> ## About sufficient contribution
>
> I still tend to think that the discovery of weights invariances influencing LMC is not a significant contribution by itself.
>
> Firstly, it is intuitive without any analysis: instead of computing the barrier between two fixed models, you are allowed to permute one of them before that. The bigger the set of permutations you consider, the higher the probability of finding a model that will have a lower barrier than the first one.
>
> Secondly, in [1] it has already been supported by extensive numerical experiments.
>
> I also want to note down, that of course, permutation search space is enormous in general, and I agree that the current paper did a good job in reducing this search space for the soft trees, but I don't think that it is enough for a paper.
>
> ### Example with ReLU
>
> I am not sure that scaling invariance for ReLU is a relevant example, because it is applied to layer outputs, not to model parameters.
>
> ## About empirical analysis
>
> In my original review, I did not mention some of the points below, because I was confused by some results and also missed some of them, for that I am sorry.
>
> - Almost no ablations are made (e.g. for the size of the ensemble).
> - In general, instead of a more detailed analysis of different cases, authors often average accuracy across all 16 datasets losing a lot of information.
>   - For example, the authors do not explain why and when the interpolated model has higher accuracy than the models it is interpolated from (e.g. for MagicTelescope it happens but for bank-marketing it does not in Figure 5) - this explanation is crucial as model merging is the main application of this paper.
>   - There is no explanation of when a barrier between models exists (e.g. it exists for Bioresponse, Higgs, eye_movement in Figure 5).
>
> ## About questionable results
>
> I find results strange in general and I am not satisfied with authors explanations for the reported numbers:
> - 2 layer MLP performs worse than 1 layer MLP in Table 2.
> - Decision list while being a version of a tree with much fewer parameters (see Figure 4) performs on par with the full version (see Table 3, 4).
>
> It shows that datasets used for evaluation are not representative, because they can be solved already by 1 layer MLP and decision list.
> That leads me to the problem of too simplistic datasets.
>
> ### Too simplistic datasets
> The selected set of datasets for experiments is different from the ones used by the soft trees community, for example, why were not Yahoo, Click and Microsoft used (see e.g. Table 5 in [2]). I think using such datasets is important for validating LMC hypothesis on a more realistic scale of at least 500K samples (in contrast to 14/16 datasets from your paper having less than 100K samples - see Table 5).
>
> I must admit that Higgs dataset was used as well as in [2], but I have two questions regarding it. Firstly, why does it have 940K samples (see Table 5) but in [2] it has 10.5M samples (see Table 5 in [2]). Secondly, whya your models exhibit accuracy of 66% (see Figure 5 for Higgs) while in the paper from 2020 they achieved 76% (see Table 1 in [2])? Does it mean that models are undertrained?
>
> ## About paper structure
>
> Even though authors considered these points as minor, I think that they are important and require a major rewriting of the paper:
> - Main parts of matching proposed in this paper are not explained in the main text: algorithms for weights and activation matching are hidden in Appendix, linear assignment problem (LAP) is not formulated at all (in [1] it was stated in eq. 1).
> - Related work section does not exist.
>
> ## Soft tree can be seen as hierarchical mixture of experts
>
> I think that it is a huge stretch. Can we say that boosting algorithms are instances of hierarchical mixtures of experts and studying them helps improve Mistral?
>
> ## Question about code
>
> According to the checklist the code is provided but I could not find it and the authors did not comment on this in the rebuttal.
>
> [1] Ainsworth et al., Git Re-Basin: Merging Models modulo Permutation Symmetries, ICLR2023
>
> [2] Popov, Sergei, Stanislav Morozov, and Artem Babenko. Neural oblivious decision ensembles for deep learning on tabular data. ICLR2020

---

> > ### Author Response · Authors · 2024-08-12
> >
> > Thank you for your detailed comments.
> >
> > > Firstly, it is intuitive without any analysis: instead of computing the barrier between two fixed models, you are allowed to permute one of them before that. The bigger the set of permutations you consider, the higher the probability of finding a model that will have a lower barrier than the first one.
> >
> > Let me present a simple counterexample. When the number of trees is fixed, a deeper perfect binary tree has a greater number of subtree flip invariance patterns. According to your intuition, it should be easier to achieve LMC with a deeper tree. However, as shown in Figure 5, this is not the case in reality.Therefore, your reasoning is unfortunately not correct, and our analysis is essential for a deeper understanding of LMC.
> >
> > > I am not sure that scaling invariance for ReLU is a relevant example, because it is applied to layer outputs, not to model parameters.
> >
> > By scaling the parameters of the layer just before applying the ReLU by a factor of $\alpha$ and scaling the parameters of the layer right after applying the ReLU by a factor of $1/\alpha$, functional equivalence is achieved through parameter adjustments.
> >
> > > Almost no ablations are made (e.g. for the size of the ensemble).
> >
> > We strongly emphasize that we have conducted an ablation study, and Figure 5 presents the results. We have investigated the changes in behavior with respect to tree depth and ensemble size. The results are also utilized in the latter part of the discussion.
> >
> > > For example, the authors do not explain why and when the interpolated model has higher accuracy than the models it is interpolated from
> >
> > > There is no explanation of when a barrier between models exists
> >
> > We believe this is an important perspective, while it is still an open problem for the community. Even when considering neural networks, there is currently no clear answer to your question. We have partially addressed this issue by evaluating LMC from a practical standpoint on tabular datasets, whereas previous research has mainly focused on datasets like MNIST and CIFAR10.
> >
> > > It shows that datasets used for evaluation are not representative, because they can be solved already by 1 layer MLP and decision list. That leads me to the problem of too simplistic datasets.
> >
> > > I must admit that Higgs dataset was used as well as in [2], but I have two questions regarding it. Firstly, why does it have 940K samples (see Table 5) but in [2] it has 10.5M samples (see Table 5 in [2]). Secondly, whya your models exhibit accuracy of 66% (see Figure 5 for Higgs) while in the paper from 2020 they achieved 76% (see Table 1 in [2])? Does it mean that models are undertrained?
> >
> > The dataset we are using is a well-known dataset used for tabular data benchmarking, known as the Tabular Benchmark [8]. During the construction of this dataset, easy datasets were deliberately avoided, and difficult datasets were used instead. Regarding the Higgs dataset, we are using a version that has been formatted by the Tabular Benchmark, so there may be some changes in the number of instances. Additionally, in terms of performance, since we conducted sampling during training according to the practices of the Tabular Benchmark, there may be a difference compared to the performance on the full dataset.
> >
> > > Soft tree can be seen as hierarchical mixture of experts. I think that it is a huge stretch. Can we say that boosting algorithms are instances of hierarchical mixtures of experts and studying them helps improve Mistral?
> >
> > The understanding that a soft tree can be interpreted as a mixture of experts is a well-known interpretation. For example, Jordan & Jacobs [7] proposed this in 1993, and it has been cited more than 4,000 times. Furthermore, since general LLMs are trained using gradient methods, considering boosting is not straightforward if the goal is to contribute to their development. Please note that since MoE refers to the model structure, it is independent of the training algorithm.
> >
> > > According to the checklist the code is provided but I could not find it and the authors did not comment on this in the rebuttal.
> >
> > Please download the supplementary material from the OpenReview platform. You can download the zip file by clicking the button located at the top of the console of this paper.
> >
> > ----
> >
> > [8] Grinsztajn et al., Why do tree based models still outperform deep learning on typical tabular data? NeurIPS 2022 Datasets and Benchmarks Track

---

> > > ### Comment · Reviewer_fRZd · 2024-08-12
> > >
> > > I thank the authors and reviewer Varw for their detailed answers and comments. After carefully reviewing them, I changed my score to 5 (borderline accept). Here is my explanation for doing that, I hope it will be helpful for authors to make their paper stronger and for reviewer JSfD to reconsider their score:
> > >
> > > ## Defeating original reasons for low score
> > >
> > > ### Soft trees are used in tabular community
> > >
> > > I originally thought soft trees were niche architecture that nobody uses in practice. After discussion with the authors I believe that the study of soft trees is well motivated as they are used in practice e.g. by [2] and [3].
> > >
> > > ### My misreading of results
> > >
> > > I missed several important experiments (e.g. ablation and model merging as application of the paper), mistakenly said that datasets are too simplistic, however, authors used a well known tabular data benchmark [2].
> > >
> > > ### Oversimplifying authors' contribution
> > >
> > > I oversimplified the author’s contribution by saying that it is intuitive without any analysis that by increasing permutation search space we can only decrease the barrier between models hence improving LMC. While in theory, it is true, in practice the search space is so big that optimal permutation is hard to find as it is an NP-hard problem [1]. Therefore, AM and WM solve it only approximately. Since nobody has provided soft tree matching algorithms before, it is not as straightforward as I originally argued.
> > >
> > > ## Remaining weaknesses that are not enough for rejection
> > >
> > > ### Absence of rigorous analysis
> > >
> > > Lack of theoretical justification and explanation of when model merging works and low barrier is achieved are indeed limitations, but the same also holds for other LMC papers e.g. for [3], therefore, I think that we should not reject this paper based on that, as it still provides valuable insights for matching soft trees.
> > >
> > > ### Insufficient explanation of key method parts
> > >
> > > I still find writing unsatisfactory, especially the explanation of Algorithms (which take an important role in method but are hidden in Appendix) and LAP formulation. However, I believe that it can be fixed in camera-ready version and not a reason for rejection.
> > >
> > > ### Strange results
> > >
> > > I still find results for MLP and linked tree strange, but that alone is not sufficient to reject the paper in my opinion:
> > >
> > > - 2 layer MLP performs worse than 1 layer MLP in Table 2.
> > > - Decision list while being a version of a tree with much fewer parameters (see Figure 4) performs on par with the full version (see Table 3, 4).
> > >
> > > [1] Ainsworth et al., Git Re-Basin: Merging Models modulo Permutation Symmetries, ICLR2023
> > >
> > > [2] Grinsztajn et al., Why do tree based models still outperform deep learning on typical tabular data? NeurIPS 2022 Datasets and Benchmarks Track
> > >
> > > [3] Popov, Sergei, Stanislav Morozov, and Artem Babenko. Neural oblivious decision ensembles for deep learning on tabular data. ICLR2020

---

> > > > ### Author Response · Authors · 2024-08-12
> > > >
> > > > Thank you for engaging in a thorough discussion on many aspects of our work, updating the score, recommending acceptance, and encouraging other reviewers to reconsider their scores. We sincerely appreciate all of these efforts. We will do our best to incorporate your valuable comments into the camera-ready version to improve the quality of our paper.

---

### Official Review · Reviewer_JSfD · 2024-07-15

**Soundness:** 3
**Presentation:** 2
**Contribution:** 3
**Rating:** 4
**Confidence:** 3

**Summary:**

This paper empirically shows that separately trained tree emsemble models can show Linear Mode Connectivity (LMC) when considering tree invariant operations.

**Strengths:**

- The exploration of LMC on tree emsemble models is interesting.
- The computational process is clearly stated which makes this paper easy to follow.

------

After reading author rebuttal and discussion with other reviewers, I decide to increase my rating of this paper to borderline reject.

**Weaknesses:**

This paper does not provide any insights into the question of LMC in neural networks, as it is exploring a totally different model. Although it is always interesting to consider LMC in another senario, I find the contribution of this paper rather insignificant and incremental, since it is basically applying the same idea of [1] to another model. I do not want to deny the author's valueable efforts in exploring symmetry in a new model and using it to achieve LMC, but I just feel that the contribution of this paper may not be sufficient for it to be accepted by this conference.

One possible direction I can suggest for the authors to enhance the current paper is, if any non-trivial theory about LMC can be made on the tree ensemble model setting, then this work will be much more exciting. The underlying reason why neural networks can be made linear connected is not yet clear, and the is hard to study due to the non-linear nature of deep NNs. If the authors can show that the tree ensemble model can be an alternative model to study LMC from a theoretical perspective, then this will make the current work more valuable and intresting.

[1] Git Re-Basin: Merging Models modulo Permutation Symmetries

**Regard writting**
The intro is kind of confusing for readers who are not familiar with the tree ensemble models. It's even unclear whether 1) it is a new model ensembling method for neural networks, or 2) it is a new model, or 3) it is a training method. Although those questions are addressed after reading the detailed definition of tree ensemble in Section 2.2, I think it is better to make it clear in the intro to avoid any confusing.

**Questions:**

See Weaknesses.

**Limitations:**

Authors discussed the limitations.

---

> ### Author Rebuttal · Authors · 2024-08-04
>
> Thank you for your review.
>
> > This paper does not provide any insights into the question of LMC in neural networks, as it is exploring a totally different model. Although it is always interesting to consider LMC in another senario, I find the contribution of this paper rather insignificant and incremental, since it is basically applying the same idea of [1] to another model. I do not want to deny the author's valueable efforts in exploring symmetry in a new model and using it to achieve LMC, but I just feel that the contribution of this paper may not be sufficient for it to be accepted by this conference.
>
> First, we would like to emphasize the importance of our research. As noted in the introduction, while soft tree ensembles are distinct from neural networks, they are also highly regarded models in their own right. Understanding their behavior is of great importance for the ML community.
>
> Our contribution is not merely the application of the results from [1] to another model. We identify the necessity of unique invariances in the context of tree ensembles and demonstrate its importance. This perspective was previously unknown in the community. Additionally, we propose an approach to modify tree structures to adjust the number of patterns of invariance required to achieve LMC, which is an essential consideration for practical applications and this idea can be applied to other model structures as well.
>
> While the existence of permutation invariance in neural networks was known, investigating whether considering this invariance could achieve LMC was a non-trivial question. Studies addressing this perspective have made significant impacts on the community [1]. For example, architectures like transformers, with their various modules such as attention mechanisms, have shown the importance of considering their unique invariances. This work has been highly regarded and accepted in a recent top-tier conference (ICLR2024) [2].
> Moreover, reviewer Varw, who has certain expertise in LMC, also praised our contribution: `Honestly, I enjoyed reading this paper. Although I am not specialized in tree ensembles, I have certain expertise in LMC, and was pleased to find that it is also relevant for DTE models. I think that this contribution is novel and significant.`. This also supports the impact our research has on the community.
>
> We believe our diverse contributions have significance that deserve the conference.
>
> > One possible direction I can suggest for the authors to enhance the current paper is, if any non-trivial theory about LMC can be made on the tree ensemble model setting, then this work will be much more exciting. The underlying reason why neural networks can be made linear connected is not yet clear, and the is hard to study due to the non-linear nature of deep NNs. If the authors can show that the tree ensemble model can be an alternative model to study LMC from a theoretical perspective, then this will make the current work more valuable and intresting.
>
> You are right that the community currently struggles to theoretically explain why LMC can be achieved in neural networks. As shown in Equation (5), soft tree ensembles and MLPs have fundamental similarities. Thus, deepening our understanding of soft tree ensembles will simultaneously lead to a better understanding of neural networks. Our contribution enables the community to consider whether soft tree ensembles can serve as an alternative model to neural networks for studying LMC, which can serve as a milestone for future research.
>
> ----
>
> [1] Ainsworth et al., Git Re-Basin: Merging Models modulo Permutation Symmetries, ICLR2023
>
> [2] Imfeld et al., Transformer Fusion with Optimal Transport, ICLR2024

---

> ### Comment · Reviewer_JSfD · 2024-08-13
>
> After reading the rebuttal provided by the authors and discussing with other reviewers, I decide to raise my rating to this paper to borderline reject, for the following reason:
> 1. Previously, I didn't think it is very meaningful to explore LMC on DTE, but that I'm not interested in DTE does not mean other researchers are not interested in it, and those who are working on DTE might feel excited about this work.
> 2. The techniques and findings of this paper can be helpful to future researchers in this field.
>
> I decide to maintain my opinion in the negative side because in my view this paper is still relatively incremental and contributes very little the core issues of LMC.

---

> > ### Author Response · Authors · 2024-08-14
> >
> > We appreciate your acknowledgment of our contribution. Regarding our work in relation to the core issues of LMC, while LMC has traditionally been studied primarily within the context of neural networks, we believe that extending these discussions to include other model architectures, as we have done in our manuscript, is both essential and nontrivial for a more comprehensive understanding of the fundamentals of LMC.
> >
> > We would like to once again express our sincere thanks for the valuable feedback and insights. We will certainly incorporate your comments to improve the quality of our camera-ready version.

---

### Official Review · Reviewer_Agya · 2024-07-17

**Soundness:** 3
**Presentation:** 3
**Contribution:** 3
**Rating:** 6
**Confidence:** 2

**Summary:**

This paper aims to achieve LMC for soft tree ensembles. Akin to achieve LMC for neural network after accounting for permutation invariance, the authors introduce three different kinds of invariance in soft tree ensembles: tree permutation invariance, subtree flip invariance, splitting order invariance. Additionally, the authors demonstrate that better LMC can be achieved after considering all three kinds of invariance.

**Strengths:**

1. The idea of extending LMC from neural networks to differentiable tree ensembles is interesting.
2. Invariances beyond permutation variance are identified for differentiable tree ensembles. The authors demonstrate the effectiveness of accounting for these invariances when doing matching.

**Weaknesses:**

1. I am not familiar with differentiable tree ensembles, therefore, I would suggest the authors put more efforts on explaining tree ensembles and illustrating the invariances.
2. Another concern is about the motivation. This study is motivated by the question "Can LMC be achieved for soft tree ensembles?" but why would we achieve LMC for the tree ensembles? I would expect more elaboration on the motivation.

**Questions:**

N/A

---

> ### Author Rebuttal · Authors · 2024-08-04
>
> Thank you for your review.
>
> > I am not familiar with differentiable tree ensembles, therefore, I would suggest the authors put more efforts on explaining tree ensembles and illustrating the invariances.
>
> Thank you for your comment. We will include an explanation with diagrams similar to Figure 1 in [1] in the Appendix of the camera-ready version. Please note that the paper is self-contained, with all definitions provided. There is already a diagram regarding invariances; please see Figure 2.
>
> > Another concern is about the motivation. This study is motivated by the question "Can LMC be achieved for soft tree ensembles?" but why would we achieve LMC for the tree ensembles? I would expect more elaboration on the motivation.
>
> As stated in the introduction section, achieving LMC justifies the non-trivial phenomenon that model training consistently succeeds despite non-convex optimization nature. Additionally, it enables the application of model merging. These theoretical and practical aspects motivate the investigation of LMC for various models including soft tree ensembles. These aspects are also highlighted as motivations in existing LMC studies like [2].
>
> The soft tree is a model used in typical supervised learning that is distinct from neural networks, particularly noted for its application to tabular datasets. Soft trees have gained attention for combining the interpretability and inductive biases of decision tree ensembles with the flexibility of neural networks. As a result, they are implemented in well-known open-source software, such as PyTorch Tabular [3], highlighting the importance of deepening our understanding of this model. We plan to add this information in the introduction of the camera-ready version.
>
> ----
>
> [1] Frosst and Hinton, Distilling a Neural Network Into a Soft Decision Tree, CEX workshop at AI*IA 2017 conference
>
> [2] Ainsworth et al., Git Re-Basin: Merging Models modulo Permutation Symmetries, ICLR2023
>
> [3] Manu Joseph, PyTorch Tabular: A Framework for Deep Learning with Tabular Data, arXiv 2104.13638

---

> > ### Comment · Reviewer_Agya · 2024-08-13
> >
> > Thank you for your detailed response and I will maintain my current score. Besides, I strongly recommend the authors to elaborate more on the motivation side in future revision.

---

> > > ### Author Response · Authors · 2024-08-13
> > >
> > > We greatly appreciate your insightful feedback and will make efforts to incorporate it into the camera-ready version to further enhance the quality of our paper.

---

### Author Rebuttal · Authors · 2024-08-04

Thank you for your valuable comments. We would like to engage in discussions by replying to each of your comments. We provide a PDF of an additional figure to address a comment from Reviewer Varw.

---

### Decision · Program_Chairs · 2024-09-25

**Decision:**

Reject

**Comment:**

The paper presents an empirical study of linear mode connectivity (LMC) in soft tree ensembles. On one hand it demonstrates that LMC exists for soft tree ensembles. On the other hand it studies invariances in soft tree ensembles and demonstrates the importance of taking these invariances into account when investigating LMC.

The paper is really borderline. There has been a long discussion between reviewers. While all reviewers agreed that existence of LMC for soft trees is interesting, the importance of this finding is limited by the fact that soft tree ensembles are model-wise related to neural networks. Moreover,  reviewers highlighted that  the paper teaches us "something about weight or activation matching (WM/AM): ...we can effectively reduce the permutations search space by manually finding permutations that result in functionally equivalent models and then consider only them during matching", which is an finding interesting to future researchers. However, the paper could be substantially  improved by making the motivation more clear (e.g. I guess you hint to the existence of multiple similar local optima but the statement that "achieving LMC justifies the non-trivial phenomenon that model training consistently succeeds despite non-convex optimization nature" is strictly speaking not correct, that is, even in models, in which LMC is observed, optimization can go wrong and even if non-convex optimizations leads to good local optima, it does not follow that LMC exists. Thus LMC and the success of non-convex optimization is not directly connected).

Despite some interesting observations made by the paper, taking the competitiveness of the venue into account, I vote for rejection.